# Symbolic Regression with a Learned Concept Library

**Arya Grayeli**[*]
UT Austin, Foundry Technologies

**Atharva Sehgal**[*]
UT Austin

**Omar Costilla-Reyes**
MIT

**Miles Cranmer**
University of Cambridge

**Swarat Chaudhuri**
UT Austin

## Abstract

We present a novel method for symbolic regression (SR), the task of searching for compact programmatic hypotheses that best explain a dataset. The problem is commonly solved using genetic algorithms; we show that we can enhance such methods by inducing a library of abstract textual concepts. Our algorithm, called LASR, uses zero-shot queries to a large language model (LLM) to discover and evolve abstract concepts occurring in known high-performing hypotheses. We discover new hypotheses using a mix of standard evolutionary steps and LLM-guided steps (obtained through zero-shot LLM queries) conditioned on discovered concepts. Once discovered, hypotheses are used in a new round of concept abstraction and evolution. We validate LASR on the Feynman equations, a popular SR benchmark, as well as a set of synthetic tasks. On these benchmarks, LASR substantially outperforms a variety of state-of-the-art SR approaches based on deep learning and evolutionary algorithms. Moreover, we show that LASR can be used to discover a new and powerful scaling law for LLMs.

## 1 Introduction

Symbolic regression (SR) [33] is the task of finding succinct programmatic hypotheses — written in a flexible, domain-specific programming language — that best explain a dataset. Initially proposed in the 1970s, SR has recently emerged as a prominent approach to automated scientific discovery, with applications in domains from astrophysics [30, 12] to chemistry [2, 22] to medicine [51].

Computational complexity is a fundamental challenge in SR, as the space of hypotheses that an SR algorithm must search is discrete and exponential. Previous work has approached this challenge using methods like genetic programming [41, 10], neural-guided search [11, 43], deep reinforcement learning [38] and hybrid algorithms [28]. However, new tools to enhance the scalability of SR remain a critical need for applications in SR and scientific discovery.

In this paper, we show that *abstraction* and *knowledge-directed discovery* can be powerful principles in building such scaling tools in SR. State-of-the-art genetic algorithms for SR [10] evolve pools of candidate hypotheses using random mutation and crossover operations. By contrast, a human scientist does not just randomly mutate their explanations of data. Instead, they synthesize background knowledge and empirical observations into abstract concepts, then use these concepts to derive new explanations. We show that zero-shot queries to large language models (LLMs) can be used to implement such a discovery process on top of a standard SR algorithm.

Concretely, we present a new method for symbolic regression, called LASR, that discovers a library of abstract, reusable and interpretable textual *concepts* and uses it to accelerate SR. LASR alternates between three phases: (i) *concept-directed hypothesis evolution*, where standard genetic operations

---

[*]Equal contribution; Correspondence to atharvas@utexas.edu

38th Conference on Neural Information Processing Systems (NeurIPS 2024).

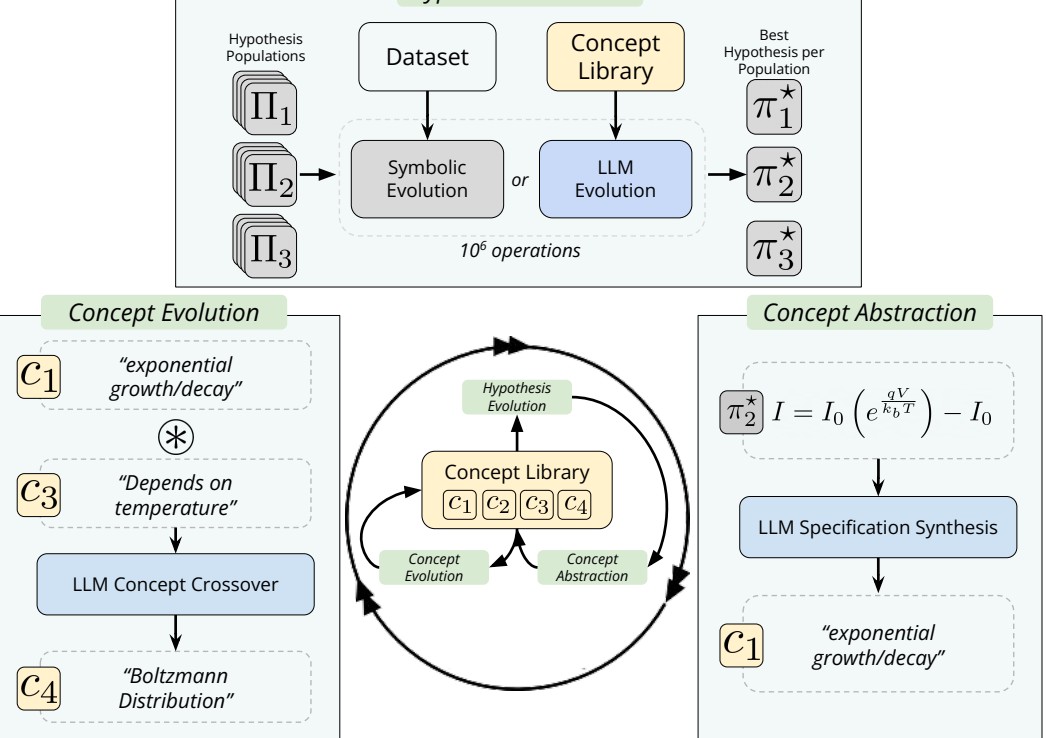

Figure 1: An overview of LASR. LASR iteratively refines a library of interpretable textual concepts which are used to bias the search for hypotheses for scientific discovery tasks. This involves three distinct phases: (**Top**) finding optimal hypotheses within a concept-directed hypothesis evolution, (**Right**) leveraging the optimal hypotheses to find new concept abstractions, and (**Left**) iterating on learned concepts to discover new concepts to accelerate hypothesis evolution. LASR introduces an orthogonal direction of improvement over current symbolic regression algorithms [10] (in gray).

over hypotheses are interleaved with LLM-guided mutation and crossover operations conditioned on known library concepts; (ii) the LLM-based *abstraction* of patterns in known high-performing hypotheses into new concepts; and (iii) the LLM-directed *evolution of concepts* into more succinct and general forms. Together, these three steps form an open-ended alternating maximization loop that combines evolutionary exploration with the exploitation of the LLM's background knowledge and in-context learning ability.

We experimentally compare LASR on Feynman Equations [26] — a popular SR benchmark in which the goal is to discover 100 equations from the Feynman Lectures in Physics — against several state-of-the-art genetic and deep learning approaches. LASR can discover 66 of the 100 target equations, while the best existing approach can solve 59. To address the concern that LASR's performance could be attributed to test set leakage, we compare LASR with a state-of-the-art genetic approach on a suite of synthetic benchmarks. We show that LASR substantially outperforms the baseline. Finally, we support the contribution of LASR to the research community by evaluating the methodology in a case study where the goal is to find new scaling laws for LLMs.

In summary, the contributions of this paper are as follows:

- We pose the problem of discovering an open-ended, reusable concept library that can accelerate solutions to downstream SR tasks.

- We present LASR, a method for combining zero-shot LLM queries and standard evolutionary operations to simultaneously induce a concept library and high-performing hypotheses. LASR's strategy of using LLMs to accelerate evolutionary algorithms may have future applications in settings beyond SR.

- We offer promising experimental results, including a demonstration that LASR outperforms state-of-the-art algorithms in standard SR tasks and synthetic domains, as well as a case study that uses LASR to discover a novel LLM scaling law.

## 2 Problem Formulation

**Symbolic Regression.**    We formulate symbolic regression (SR) as a program synthesis [6] problem. The inputs to this problem include a language $\mathcal{L}$ of programmatic hypotheses and a dataset $\mathcal{D} := \{(\mathbf{x}_i, \mathbf{y}_i)\}_{i=1}^N$ of input-output examples. The syntax of $\mathcal{L}$ is described by a *context-free grammar* [24]. The grammar allows each hypothesis $\pi$ to be represented using a set of mathematical operators (e.g., addition, multiplication, trigonometric functions) that facilitate the composition of simpler hypotheses into more complex ones. We abstractly define the *fitness* of a hypothesis $\pi$ as the likelihood $p_\mathcal{L}(\mathcal{D} \mid \pi)$ that it generates $\mathcal{D}$.

In order to prevent finding non-useful solutions, we impose a *prior probability distribution* $p_\mathcal{L}(\pi)$ over hypotheses $\pi$ that penalizes syntactically complex hypotheses. We now pose SR as the task of finding a hypothesis $\pi^\star$ that maximizes the fitness while minimizing syntactic complexity. The problem can be expressed as a maximum a posteriori (MAP) estimation problem [15]:

$$\pi^\star = \arg\max_\pi p_\mathcal{L}(\pi|\mathcal{D}) = \arg\max_\pi \underbrace{p_\mathcal{L}(\mathcal{D}|\pi)}_{\text{optimization}} \cdot \underbrace{p_\mathcal{L}(\pi)}_{\text{regularization}} \tag{1}$$

Recent work leverages large language models (LLMs) for program synthesis [8, 19, 32]. Large language models (LLMs) approach program synthesis as a token prediction problem, directly approximating the likelihood of programs by training on internet-scale datasets. That is,

$$p_\mathcal{L}(\pi|\mathcal{D}) \approx p_{\text{LLM}}(\langle\pi\rangle \mid \langle\mathcal{L}\rangle, \texttt{desc}(\mathcal{D})), \tag{2}$$

where $\langle\pi\rangle$ and $\langle\mathcal{L}\rangle$ are, respectively, textual representations of $\pi$ and a specification of the syntax of $\mathcal{L}$, and the *task description* $\texttt{desc}(\mathcal{D})$ is a few-shot serialization of a subset of the examples in $\mathcal{D}$.

**Symbolic Regression with Latent Concept Libraries.**    Classical symbolic regression typically assumes no prior knowledge or intuition about the problem. In contrast, human scientific discovery often leverages empirical patterns [52] and intuitions derived from previously observed data. For example, recognizing a 'power law relationship between variables' has led to the formulation of fundamental empirical laws across various fields, such as the Arrhenius equation in Chemistry, the Rydberg formula in Physics, Zipf's law in Linguistics, and Moore's law in Computer Science.

We model such empirical patterns as natural-language *concepts* drawn from a latent *concept library* $\mathcal{C}$. We frame the relationship between the concept library and programs as a Hierarchical Bayesian model consisting of: (i) a *prior* $p(\mathcal{C})$ representing the natural distribution over concept libraries; (ii) a model $p_\mathcal{L}(\pi \mid \mathcal{C})$ that quantifies the likelihood of various hypotheses for a given concept library $\mathcal{C}$; and (iii) the previously mentioned fitness function $p_\mathcal{L}(\mathcal{D} \mid \pi)$ for programs $\pi$. We assume that the distributions $p(\mathcal{C})$ and $p_\mathcal{L}(\pi \mid \mathcal{C})$ can be approximated using LLMs. That is, we can prompt an LLM to generate interesting concepts, and we can prompt an LLM with a set of concepts to generate token-sequence representations of hypotheses that adhere to the concepts. Now we state the problem of *symbolic regression with latent concept learning* as one of simultaneously inducing an optimal concept library and an optimal programmatic hypothesis:

$$\arg\max_{\pi,\mathcal{C}} p(\pi, \mathcal{C}|\mathcal{D}) = \arg\max_{\pi,\mathcal{C}} \underbrace{p(\mathcal{D}|\pi)}_{\text{By execution}} \cdot \underbrace{p(\pi|\mathcal{C})}_{\text{By LLM}} \cdot \underbrace{p(\mathcal{C})}_{\text{By LLM}} \tag{3}$$

## 3 Method

LASR performs a two-stage evolution over natural-language concepts and programmatic hypotheses. The two stages follow an alternating maximization strategy shown in Figure 1: (1) *Hypothesis evolution*: We fix the set of concepts and focus on maximizing the hypotheses' fitness to the dataset, and (2) *Concept abstraction and evolution*: We leverage the best hypotheses found to induce a new library of concepts.

In the rest of this section, we first describe PySR, the SR algorithm [10] that LASR extends. Next, we show how to modify this algorithm into one guided by natural-language concepts. Finally, we show how these concepts can be naturally extracted and evolved into new concepts. The full LASR algorithm is presented in Algorithm 1 and visualized in Figure 2. LASR is built in Julia with an additional Python interface[2] and uses an open-source, optimized framework for LLM inference [25].

---

[2]Artifacts available at `https://trishullab.github.io/lasr-web`

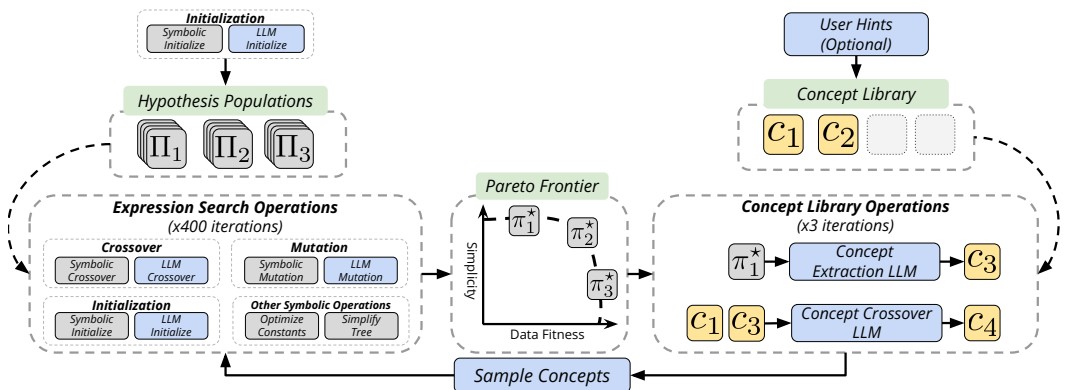

Figure 2: A single step of LASR. LASR induces multiple hypothesis populations that are evolved using a scalable evolutionary algorithm. Concept guidance is provided by randomly replacing symbolic operations with concept-directed LLM operations with probability $p$. After each iteration, the top-performing programs are summarized into natural language concepts, which are evolved to form new concepts that are sampled to guide the search in the next iteration.

**Base Algorithm: PySR.** LASR builds on PySR [10], a scalable, parallelizable genetic search algorithm for SR. The search in PySR maintains multiple populations $\{\Pi_1, \ldots, \Pi_k\}$ of hypotheses, with each hypothesis represented as an expression tree. In its *initialization* step, captured by a procedure INITIALIZEPOPULATIONS, PySR creates a new expression at random to insert into a population. After running this step, PySR runs a genetic search, encapsulated in a procedure SRCYCLE, which evolve these populations in parallel, simplifies and optimizes the constants of the resulting hypotheses, and then migrates top-performing hypotheses between populations.

Like other evolutionary algorithms, the search in PySR uses symbolic *mutation* and *crossover* operations. The mutation step is broken into many categories, each with distinct weighting, to either mutate a constant, mutate an operator, add a node (append, prepend, insert), delete a subtree of an expression tree, simplify the tree, initialize a new tree, or do nothing. One of these operations is randomly selected at each call to a mutation request, and each operation executes itself at random but within user-provided constraints. For example, deleting a subtree is done by choosing a random node to replace with a randomly-generated leaf node such as a feature or constant. The crossover step involves swapping random subtrees of two expressions in a population.

**LLM-guided Hypothesis Evolution.** LASR speeds up PySR by injecting natural language priors into its search procedure. To do this, we modify the INITIALIZEPOPULATIONS procedure to use an LLM-augmented initialization operation, and the SRCYCLE routine to use LLM-augmented versions of its symbolic mutation and crossover operations. The altered procedures are named LLMINIT, LLMMUTATE, and LLMCROSSOVER, respectively. These operations do not *replace* their standard genetic counterparts. Instead, we introduce a hyperparameter $p$ that, with a fixed probability, substitutes the standard genetic operation with the LLM-based operation. This enables "doping" each population with a program that respects the language priors, while ensuring that we do not bottleneck the local exploration of the search space.

The LLM-guided operations follow the same base format: they sample multiple concepts from the concept library, concatenate these concepts with the task-specific variable names and language operations, and append a specialized prompt for each task. We employ zero-shot prompts (see Appendix A.2 for more details) to avoid sampling biases. In further detail:

- LLMINIT: The LLMINIT function takes an initial set of concepts and uses them to initialize the populations for the evolutionary search step. The initial set of concepts can either be instantiated from an optional set of user-provided "hints" or generated by the LLM.

- LLMMUTATE: For mutation within a population, we sample a set of $l$ concepts from the concept library $C$, and construct a prompt that uses this set of concepts to mutate an expression $\pi_i$ into $\pi_j$. The prompt to the LLM takes inspiration from the standard genetic mutation operation, and asks it to mutate the expression given the concepts sampled from the library.

**Algorithm 1** Pseudocode for LASR. LASR takes as input an optional set of user-provided hints $\mathcal{C}_0$, a dataset of input-output pairs of high-dimensional data $\mathcal{D}$, and four hyperparameters: the number of iterations $I$, the number of populations $K$, the number of steps for concept evolution $M$, and the mixture probability of using LLM-based or GP-based evolutionary operators $p$. LASR produces two artifacts: the evolved library of concepts $\mathcal{C}$ and the expression with the highest fitness score $\pi^\star$.

```
 1: function LASR(C₀, D = {(xᵢ, yᵢ)}ᵢ₌₁ᴺ, I, K, M, p)
 2:     C ← INITIALIZECONTEXTLIBRARY(C₀)                          ▷ Add (optional) user hints to library.
 3:     {Π₁, ... Π_K} ← INITIALIZEPOPULATIONS(C, K)
 4:     for _ in range(I) do
 5:         for i in range(K) do
 6:             Πᵢ ← SRCYCLE(Πᵢ, D, C, p)                         ▷ Interleaved Symbolic + LLM Search
 7:         F ← EXTRACTPARETOFRONTIER({Π₁ ... Π_K}, D)            ▷ Includes positive + negative programs
 8:         C ← C ∪ CONCEPTABSTRACTION(F, C)
 9:         for _ in range(M) do
10:             C ← CONCEPTEVOLUTION(C)
11:     π* ← BESTEXPRESSION(F)                                    ▷ Based on dataset loss and program complexity
12:     return C, π*
```

- LLMCROSSOVER: The LLMCROSSOVER function also samples a set of $l$ concepts from the concept library along with two hypotheses $\pi_i$ and $\pi_j$ to construct a new expression $\pi_k$, which reuses sub-expression trees from the two hypotheses while respecting the sampled concepts. Our implementation is inspired by prior work [40] — see Figure 4.

**Concept Abstraction.** After each iteration of symbolic regression, we use a function EXTRACT-PARETOFRONTIER to collect: (i) the hypotheses, across all populations, that are Pareto-optimal with respect to the criteria of syntactic simplicity and dataset loss; (ii) the hypotheses with the worst loss across all populations. The resulting set of hypotheses $\mathcal{F} = \{\pi_1^\star., \pi_a^\star ... \pi_1^-., \pi_b^-\}$ captures the trends that were most helpful and most detrimental to performance during hypothesis search. Now we use the CONCEPTABSTRACTION function, which uses a zero-shot prompt to extract a natural language concept $c^\star$ that summarizes the positive trends while eschewing negative trends. This concept is subsequently added to the concept library. The prompt for the function is presented in Figure 5.

**Concept Evolution.** Each concept in $\mathcal{C}$ represents trends that were useful at a previous state in the search process. After adding new concepts into the library, we use a function CONCEPTEVOLUTION to evolve the library to include new ideas that logically follow from the ideas in the current library. The implementation of this function follows that of the LLMCROSSOVER operation in that we are using multiple concepts as a reference to generate new ones, with the key distinction that, unlike in the LLMCROSSOVER operation, the fitness of each generated concept here is difficult to quantify. Thus, we include all the generated responses in the concept library. While these concepts may sometimes be inaccurate, they increase the evolutionary algorithm's exploration ability.

## 4   Experiments

We demonstrate the effectiveness of LASR on multiple tasks integral to scientific discovery. First, we evaluate LASR's performance on the Feynman Equation dataset, a widely adopted scientific discovery benchmark, under a variety of ablations and additional priors. Second, we measure the effect of data leakage by evaluating LASR's performance on a procedurally generated synthetic dataset of challenging equations. Finally, we conduct a case study using LASR to discover LLM scaling laws with data from the BIG-Bench evaluation suite [16].

LASR's main focus is to serve as a practical toolkit for scientists. Therefore, our evaluation primarily targets slightly noisy environments, using exact solution rate to gauge performance rather than statistical similarity measures like correlation $R^2$, which are less relevant to scientific discovery applications. Additional experiments characterizing the practical behavior of LASR are in Appendix A.5. Information regarding compute usage is in Appendix A.3.1.

| GPlearn | AFP | AFP-FE | DSR | uDSR | AIFeynman | PySR | LaSR |
|---------|-----|--------|-----|------|-----------|------|------|
| 20/100 | 24/100 | 26/100 | 23/100 | 40/100 | 38/100 | 59/100 | **72/100** |

Table 1: Results on 100 Feynman equations from [49]. We report exact match solve rate for all models. LaSR achieves the best exact match solve rate using the same hyperparameters as PySR.

| Type of Solve | PySR | LaSR (Llama3-8B) | | | LaSR (GPT-3.5) |
|---------------|------|------------------|---|---|----------------|
| | | $p = 1\%$ | $p = 5\%$ | $p = 10\%$ | $p = 1\%$ |
| Exact Solve | 59/100 | 67/100 | 69/100 | 71/100 | 72/100 |
| Almost Solve | 7/100 | 5/100 | 6/100 | 2/100 | 3/100 |
| Close | 16/100 | 9/100 | 12/100 | 12/100 | 10/100 |
| Not Close | 18/100 | 19/100 | 13/100 | 16/100 | 15/100 |

Table 2: Evaluation results on Feynman dataset by cascading LaSR's LLM backbone (llama3-8b, gpt-3.5-turbo) and changing the probability of calling the model ($p$ = [0.01, 0.05, 0.10]) in the order of increasing concept guidance. LaSR outperforms PySR even with minimal concept guidance using an open-source LLM.

## 4.1 Comparison against baselines in the Feynman Equation Dataset

**Dataset**: The Feynman Equations dataset is a well established benchmark for Symbolic Regression algorithms [49]. This dataset consists of 100 physics equations extracted from the Feynman lectures on Physics. We compare against performance reported on SRBench [26]: a continuously updated public benchmark of SR methods on many datasets. Specifically, we compare against GPlearn, AFP, AFP-FE, DSR, uDSR, PySR, and AI Feynman [42, 47, 49, 28, 38]. Within this subset PySR represents an ablation of our model without the LLM genetic operations and the concept evolution (Section 3). We evaluate on a slightly noisy version of this dataset in order to simulate experimental errors common in scientific discovery domains. More details are presented in Appendix A.4.1.

**Setup**: We instantiate LaSR using `gpt-3.5-turbo-0125` [4] as the backbone LLM and calling it with $p = 0.01$ for 40 iterations, and compare our results with PySR which uses the same default hyperparameters. For the other baselines, we use the numbers reported in SRBench with one exception being uDSR [28], for which we couldn't find any benchmarking numbers. For this method, we derive the exact solve rate from [37].

**Results**: We showcase results in Table 1. We draw three observations from this experiment. First, LaSR achieves a higher exact solve rate than all other baselines. Second, both PySR and LaSR outperform the other baselines by a wide margin, indicating that scalable and efficient synthesis is imperative to practical scientific discovery algorithms. Finally, and most notably, a subset of the equations LaSR finds could not be derived with any of the previous methods.

## 4.2 Cascading Experiments

LaSR's performance is inherently bottlenecked by the reasoning capabilities of the backbone LLMs and the frequency of their invocation in each iteration. To evaluate the effect of the backbone LLM on LaSR's performance, we instantiate a model cascade over two of LaSR's hyperparameters: the backbone model (`llama3-8b` [17], `gpt-3.5-turbo-0125`) and the probability $p$ with which we call that model in the evolution step ($p$ = [1%, 5%, 10%]).

**Setup**: Our cascade operates as a tournament. We start LaSR with the configuration that provides the least language guidance (`llama3-8b` at $p = 1\%$) and progressively increase the value of $p$ and then the backbone model. Each subsequent model is only evaluated on the problems that the previous model could not solve. We compare this against PySR's performance on the Feynman equation dataset. To ensure a fair comparison, we cascade PySR using the same procedure but find it does not solve any additional equations. For this experiment, we tag each equation with a qualitative rating comparing the equation to the ground truth form (Exact Solve, Almost Solve, Close, and Not Close). An in-depth discussion on this metric is presented in Section A.7.

**Results**: Our results are presented in 2. We draw two key observations from these results. First, LaSR outperforms PySR even with minimal concept guidance (`llama3-8b` at $p = 1\%$). Second,

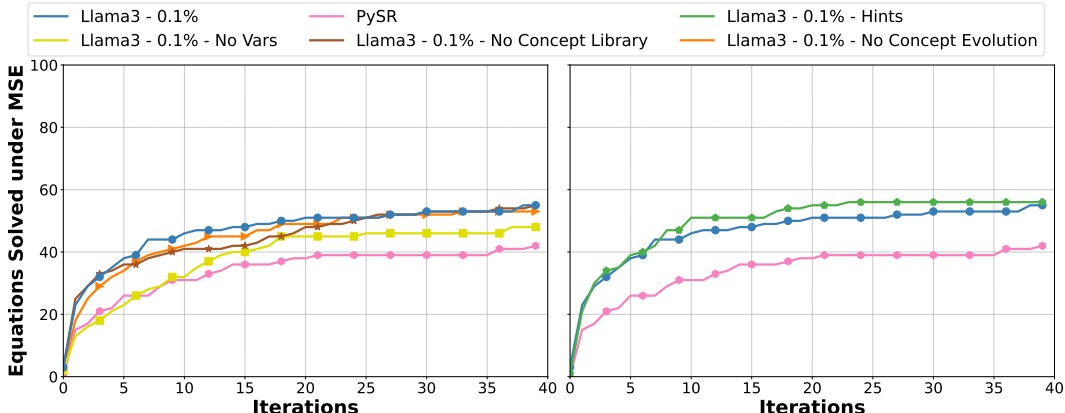

Figure 3: Evaluation results for ablations/extensions of LASR. (**Left**): We ablate three components of LASR: Concept Evolution, Concept Library, and variable names and evaluate their MSE solve rate performance on the Feynman dataset over 40 iterations. We find that each component contributes to accelerating search at different stages in the search process. (**Right**): We extend LASR by providing an initial concept library $\mathcal{C}_0$ in the form of user provided hints. We find that natural language hints significantly increases the speed of solving equations.

increasing the backbone model size and the mixture probability substantially enhances LASR's performance, indicating that as the language reasoning capabilities of LLMs improve, so will our performance.

### 4.3 Ablation Experiments

We conduct ablations on the use of Concept Evolution (skip phase three from Figure 1), Concept Library (skip phase two and three), variable names, and user hints. Figure 3 shows how these ablations affect performance over 40 iterations. We designate an equation as "solved" if, after $N$ iterations, the MSE of our predicted equation is less than $10^{-11}$. This metric differs from 'Exact Solved' as defined in the prior experiments: an equation can be 'exactly solved' yet have an MSE higher than $10^{-11}$ due to the noise floor in the target variables, and an equation can have low loss but not be an exact match. We observe from the results that: (1) Removing variable names results in a substantial performance drop, as we lose semantic meaning provided by variables (for instance, observing $\theta$ could suggest employing trigonometric functions on $\theta$). (2) Learning a concept library enables faster convergence to solutions. Without the concept library, task convergence is slower, and widens under higher concept guidance conditions ($p > 0.1\%$).

### 4.4 Qualitative Analysis and User Hints

The concept library provides an interpretable window into our evolutionary search process. To showcase the concepts learned by LASR, we take a sample equation from the Feynman dataset, the electric field of a dipole $E_f = \frac{3p_d \cos\theta \sin\theta}{4\pi\epsilon r^3}$ and comment on the libraries learned at various intervals. We see rudimentary concepts emerge in the second iteration:

*"The presence of basic trigonometric functions like sin in the good expressions contributes to their quality, indicating a connection to physical concepts such as waveforms or periodic phenomena."*

And, in subsequent iterations, the concepts become even more refined:

*"The good mathematical expressions exhibit a balance between mathematical operations such as multiplication, division, and trigonometric functions, which are known to have physical interpretations and relevance in various scientific phenomena."*

This iterative refinement helps LASR consistently maintain high-quality concepts, allowing it to converge to an exact match within 40 iterations. By contrast, PySR and the concept library ablations fail to converge on an exact match solution, returning equations that — while low-loss — involve many extra terms absent from the ground truth. This reinforces our hypothesis that injecting semantic meaning into the search process not only improves search efficiency, but also regularizes against

complex equations — as the LLM-generated concepts help filter out irrelevant terms. A deeper qualitative analysis is in Appendix A.8.

**Extending LASR with Hints**: A benefit of LASR is that its search can be initialized with a set of user-specified, natural-language "hints." To evaluate this capability, we generate hints for each equation based on variations of the chapter title of the Feynman lecture that the equation belongs to. We intentionally keep the hints vague to see if knowledge about just the general field is sufficient in improving LASR's performance. We showcase results in Figure 3. We observe a noticeable boost in performance from injecting these hints, even for our weakest performing model, indicating that even minimal user input can substantially enhance LASR's effectiveness in discovering equations.

## 4.5 Data Leakage Validation

An important consideration in using LLMs for existing SR problems is the possibility that the LLM was exposed to the hold-out problems in the validation set, presenting an unfair advantage to LLMs trained on massive datasets. Intuitively, LASR generates its own concepts which are conditioned on suboptimal programs, which are unlikely to be within the LLM's memorized responses. To validate this, we generate a dataset of 41 synthetic equations that are engineered to deviate from common physical and mathematical structures and have arbitrary variables. For example, one such equation is $y = \frac{0.782x_3+0.536}{x_2 e^{x_1}(\log x_2 - x_2 e^{\cos x_1})}$. We find that PySR struggles to solve equations with these characteristics (given 400 iterations). Hence, solving such equations hinges on the language guidance components.

We run LASR with Llama3-8B at 0.1%. We then compare our synthesized program's test set $R^2$ with that of PySR's. We justify using correlation instead of exact-match as we are not motivated by the application of results for scientific discovery in this experiment. Our results are summarized in Table 3 and show that LASR's concept-guided synthesis still provides a considerable performance boost compared to PySR – demonstrating that LASR can outperform PySR even when data leaking is not possible.

| PySR | LaSR (Llama3-8B, 0.1%) |
|---|---|
| 0.070 | **0.913** |

Table 3: Evaluation results of data leakage. We present the test set $R^2$ of PySR and of LASR on a synthetic symbolic regression dataset. Higher $R^2$ is better.

## 4.6 Using LASR to discover LLM Scaling Laws

So far, we have demonstrated that LASR can discover equations that are practical but already known (Feynman Dataset) and equations that are novel but aren't practical (Synthetic Dataset). To investigate LASR's utility in finding novel and practical empirical trends, we investigate whether LASR can discover novel LLM scaling laws on the BigBench dataset [16]. More details on this experiment are presented in Section A.6.

Traditionally, to identify an LLM scaling law, practitioners must first manually posit a "skeleton equation" with a fixed set of known variables and unknown free parameters, and then optimize the unknown parameters based on a dataset of model hyperparameters and resulting dataset fitness [23, 1, 5]. Instead of starting with a predefined equation, we use LASR to discover the skeleton equation that best fits various subsets of the BigBench dataset.

**Setup.** BigBench contains 204 tasks with scored responses from 55 language models trained with different hyperparameters. We evaluate on the subset of tasks where the preferred metric is 'Multiple choice grade' (53,812 samples). Our goal is to find the equation that best predicts the test score given the model hyperparameters and the dataset hyperparameters. We run LASR with 3840 populations of 200 candidates each for 7 hours (overnight). The runtime of LASR is comparable to other SR algorithms for this experiment as the slowest operation isn't generating candidate equations but rather optimizing and evaluating candidate equations.

**Results.** LASR discovers the following scaling law on the subset of BigBench:

$$\texttt{score} = \frac{A}{\left(\frac{\texttt{train\_steps}}{B}\right)^{\texttt{\#shots}}} + E \tag{4}$$

| Scaling Law Skeleton | MSE Loss | Free Parameters |
|---|---|---|
| Equation 4 | $0.03598 \pm 0.00265$ | 3 |
| Chinchilla [23] | $0.03610 \pm 0.00268$ | 5 |
| Modified Chinchilla | $\mathbf{0.03598 \pm 0.00264}$ | 5 |
| Residual Term Only | $0.09324 \pm 0.01992$ | 1 |

Table 4: Preliminary results on evaluating different LLM scaling laws. We measure MSE loss on a held out subset of BigBench [16]. The equation discovered with LASR performs as well as the Chinchilla equation [23] on BigBench while using less free parameters. The residual term skeleton equation ($\texttt{score} = E$) also performs well.

where $\texttt{score}$ is the MCQ grade, $\texttt{train\_steps}$ is the number of training steps for the model, and $\texttt{\#shots}$ is the number of in-context examples provided during inference. The fitted free parameters are presented in Equation 5.

*Qualitative Evaluation:* Equation 4 describes an empirical relationship between training hyperparameters (training steps) and inference hyperparameters (number of shots). It asserts that increasing the number of shots exponentially increases the model's performance for low-resource models, while having diminishing gains as the number of training steps of the model increase. This observation is consistent with work in scaling test-time compute [46].

As the output artifacts of LASR are interpretable, we can integrate this empirical relationship between training steps and number of shots into known scaling laws. Specifically, we can augment the chinchilla scaling law as follows:

$$\texttt{score} = \frac{A}{(\texttt{train\_steps} \cdot \texttt{batch\_size})^{\alpha}} + \frac{B}{(\texttt{\#params})^{\beta}} + E \qquad \text{(Chinchilla [23])}$$

$$\texttt{score} = \frac{A}{(\texttt{train\_steps} \cdot \texttt{batch\_size})^{\alpha \cdot \texttt{\#shots}}} + \frac{B}{(\texttt{\#params})^{\beta}} + E \quad \text{(Modified Chinchilla)}$$

*Quantitative Evaluation*: We fit the free parameters of each equation on the training set ($43,049$ samples) and measure the MSE loss between the actual grade and the predicted grade on the validation set ($10,763$ samples). The results are presented in Table 4. We find that the Equation 4's performance, as well as modified Chinchilla's performance, is competitive with that of Chinchilla's in predicting the MCQ grade. However, the horizontal line $\texttt{score} = E$ demonstrates acceptable performance as well. We believe increasing the scale of these preliminary experiments (with richer datasets or longer search horizon) will lead to additional empirical findings.

## 5 Related Work

**Symbolic Regression.** The field of SR started in the 1970s [18, 29] and has recently become a prominent approach to AI-for-science [33, 34, 40]. Two algorithmic themes here are:

*Non-parametric Algorithms*: Most work on SR focuses on improving search efficiency using heuristics or parallelization. Specifically, PySR [10] builds a multi-population evolutionary algorithm that incorporates various preexisting heuristics [39], and introduces novel ones such as simulated annealing, an evolve-simplify-optimize loop, and an adaptive parsimony metric. PySR has been successfully applied to study problems in domains such as cosmology [12], international economics [50], and climate modeling [20]. LASR extends PySR to enable the discovery of latent concepts.

*Parametric Algorithms*: Recent work in SR and program synthesis has often used neural networks to accelerate search [43, 40, 38, 28, 34, 13, 35]. The interplay between the neural and the symbolic components in these works can be abstracted into two categories: (1) leveraging LLMs to induce program scaffolds [34, 40, 35], and (2) learning a neural policy to accelerate search [38, 28, 43, 13]. We highlight two methods from the first category: Funsearch [40] and LLM-SR [45]. Funsearch [40] uses a pretrained LLM to implement a mutation operator on a database of executable programs under a fixed specification to find super-optimized programs in extremal combinatorics. LASR is a generalization of FunSearch: while FunSearch conditions program generation on a static "specification" (analogous to our concept library), we discover the concept library in the course of the

algorithm. We do not compare against FunSearch due to resource constraints. As for LLM-SR [45], it leverages a pretrained LLM for generating program sketches [36]. The sketch parameters are optimized and cached in a database, which is in turn used to generate new sketches. Our work is an orthogonal direction of improvement. It is technically possible to "plug" the LLM-SR framework (or other LLM-based search algorithms [35]) into LASR and use our generated concepts to guide the lower-level search component.

The second category includes methods like DSR [38], which, just like LASR, frame SR as a sequence modeling problem. However, the search in LASR leverages a learned concept library and the language and code biases in LLMs, instead of relying on amortization alone.

**Program Synthesis with Foundation Models.**   Recent work in program synthesis models program generation as a sequence prediction problem. Under this paradigm, the DSL and the input-output specification is serialized in the prompt and a code-generation foundation model [31, 7, 4] is leveraged to autoregressively generate candidate programs. This approach has been impactful in many areas including spreadsheet formula prediction [13, 8], competitive programming [32], and visual programming [48, 21, 9]. LASR is similar to work in this area in that the LLM Mutate, LLM Crossover, and LLM Initialization functions all follow the sequence prediction paradigm to synthesize mathematical equations, relying on guidance from the concept library.

**Program Synthesis with Library Learning.**   Deploying classical program synthesizers in a new domain often necessitate hand-engineering DSLs to enable scalable synthesis. This severely limits the generality and practicality of such methods. An emerging direction of research – called library learning – attempts to learn the DSL and the programs simultaneously [15, 3, 19, 27, 53, 14, 44, 54]. This is typically framed as a hierarchical Bayesian optimization problem over the space of programs and the space of library functions that generate those programs. Notably, [19] uses LLM guidance to assist in program induction and in auto-documenting learned library modules and [53] considers learning programs under a latent distribution over the space of natural language and the space of the DSL. LASR shares a similar problem formulation to these works, but optimizes over the space of programs and over the space of natural language descriptions of these programs.

## 6   Conclusion

We have presented LASR, a framework that uses zero-shot queries to an LLM to induce abstract, reusable concepts that can be used to accelerate SR. We have shown that LASR outperforms state-of-the-art approaches on the standard Feynman equation task. We have also used the algorithm to discover a novel scaling law for LLMs.

A key benefit of LASR is that its capabilities are ultimately bottlenecked by those of the underlying LLM. LLMs are rapidly gaining capability and getting cheaper, and future versions of LASR should be able to tap into this progress.

Many directions of research remain open. First, our strategy of accelerating evolutionary search with LLM-based concept induction may be applicable beyond the SR setting. Future research should explore such applications. Second, while our approach here was entirely based on in-context learning, it is worth exploring if finetuning improves the performance of the LLM. Finally, we evaluated the learned concept library exclusively on the downstream SR task. However, the library may also be valuable in other tasks such as clustering or explanation synthesis. Exploring these other tasks is an attractive topic for future work.

**Limitations.**   The current instantiation of LASR has several limitations. First, it cannot guarantee that the concepts it learns are correct or insightful. Even a concept that leads to strong performance in downstream SR tasks may do so because of quirks of the model and data, and end up misleading scientists using the method in a discovery process. Also, we do not currently have a way to ensure that the learned concepts are mutually consistent. Finally, our evaluation here was constrained by our compute budgets for LLMs and search. Whether the trends we see generalize to higher-compute regimes remains to be seen.

**Acknowledgements:** This research was partially supported by the NSF Expeditions in Computing Award #CCF-1918651, the NSF National AI Institute for Foundations of Machine Learning (IFML), and ARO award #W911NF-21-1-0009. We thank Foundry Technologies for providing substantial computational resources for our experiments.

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

# A  Appendix

## A.1  Broader Societal Impacts

We have presented LASR: a symbolic regression framework that leverages concept guidance to accelerate symbolic regression. We hope that LASR helps accelerate the search for empirical laws in the broader scientific community. In this section, we discuss the broader societal impacts and ethical considerations of our work.

*Potential for Misuse*: As with other ML techniques, symbolic regression can be leveraged by bad actors to inflict societal harm. Our experiments show that LASR accelerates the search for empirical laws from raw observations. In our setting, we are restricted to observations about physical phenomena. However, a malicious actor could misuse LASR to find patterns in datasets that violate personal rights.

*Privacy Concerns*: As mentioned before, LASR enables finding patterns in raw observations. We hope that LASR is leveraged by scientists to explain physical phenomena. However, it is possible to use such models to learn behavioral profiles without the active knowledge or explicit consent of the subjects.

*Bias and Fairness*: LASR generates two artifacts: a hypothesis that maximizes a fitness function (represented as an equation) and a library of concepts that helped discover that hypothesis. LASR ensures fairness and lack of bias in the generated equation as long as the fitness function is free of biases as well. However, we leverage foundation models to induce our library of concepts which could be trained on biased data which may reflect in our concept library. Furthermore, we cannot directly evaluate the efficacy of the concept library and its factual correctness. This doesn't affect equation generation – since equations are quantitatively evaluated. However, a human analyzing the concepts LASR learns might misinterpret trends that the model picks up on.

## A.2  LLM Prompts

Note that in the prompts in Figure 4, Figure 5, and Figure 6, we refer to our hypothesis as expressions and the concepts as hypotheses and suggestions. This prompting style was found to work best for the LLM.

## A.3  Implementation Details

### A.3.1  Compute Usage

We run all experiments on a server node with 8xA100 GPUs with 80 GB of VRAM each. However, our experiments can be reproduced with a GPU with 16 GB of VRAM. We were even able to run LASR on a laptop utilizing a quantized model hosted locally [3]. Moreover, certain models are hosted on external servers (such as `gpt-3-turbo-0125`) which allows running LASR on machines without GPUs. For this project, we chose to run `llama3-8b` using vLLM [25]. However, our framework is compatible with any LLM inference framework that allows hosting an OpenAI compliant RESTful server. For reference, each iteration makes around $60,000$ calls. Each call to the LLM is just under $1000$ tokens. This gives an upper bound on total compute of $60,000,000$ tokens per iteration if $p = 100\%$. Hence, running our model at $p = 1\%$ for 40 iterations would result in just under 25M tokens for each equation (around 4 hours on a single A100).

### A.3.2  Concept Sampling

In order to determine which concepts from the concept library we sample for the LLM Hypothesis Evolution, we randomly choose the top-K most recent concepts in the library. This ensures that we use the latest concepts, which are generally reflective of more informed hypotheses, and thus better to use. In practice, we set $K = 20$. Additionally, for Concept Evolution, we exclude the top-K most recent concepts from being used, and rather use older concepts. This is motivated by the desire to not have the concept library converge on a few ideas, rather we want diversity of thought. Our concepts are intended to be longer lasting than the hypotheses that generated them, similar to how observational data comes and goes, but the conclusions from them are more persistent.

---

[3]`TheBloke/Mistral-7B-Instruct-v0.2-GGUF` using `llama.cpp`

**(System)**
Header

You are a helpful assistant that recombines two mathematical expressions by following a few provided suggestions. You will be given three suggestions and two expressions to recombine.

An expression must consist of the following variables: `{{variables}}`. All constants will be represented with the symbol C. Each expression will only use these operators: `{{operators}}`.

**(User)**
Crossover
Prompt

Suggestion 1: `{{assump1}}`
Suggestion 2: `{{assump2}}`
Suggestion 3: `{{assump3}}`
Expression 1: `{{expr1}}`
Expression 2: `{{expr2}}`

Propose `{{N}}` expressions that would be appropriate given the suggestions and expressions. Provide short commentary for each of your decisions. End with a JSON list that enumerates the proposed expressions following this format:

**(User)**
JSON
Formatting
Instructions

```json
["expr1",
 "expr2",
 ...
 "expr{{N}}"
]
```

**(Assistant)**
Generated
Explanation

Based on the provided suggestions and expressions, I propose the following five recombined expressions:

1. `log(exp(C - sqrt(theta ^ C)) + C)`: This expression combines the exponential function from Expression 2 with the logarithmic function from Expression 1. The `- sqrt(theta ^ C)` term is taken from Expression 2 and added to the exponential function, while the `+ C` term is taken from Expression 1.

2. `(sin(C - sqrt(theta)) * (theta ^ log(theta / theta)) + C)`: This expression combines the sine function from Expression 1 with the power operation from the same expression. The `(theta ^ log(theta / theta))` term is taken from Expression 1, and the `sin(C - sqrt(theta))` term is also from Expression 1. The `+ C` term is added at the end.

3. `exp(C - sqrt(theta)) * sin(C - sqrt(theta))`: This expression combines the exponential function from Expression 2 with the sine function from Expression 1. Both the exponential and sine functions are taken from separate expressions and multiplied together.

4. `log(theta ^ log(theta / theta)) + C - sqrt(theta ^ C)`: This expression combines the logarithmic function from Expression 1 with the power operation from the same expression. The `(theta ^ log(theta / theta))` term is taken from Expression 1, and the `- sqrt(theta ^ C)` term is taken from Expression 2. The `+ C` term is added at the end.

5. `(sin(C - sqrt(theta)) ^ (theta ^ log(theta / theta)) + C)`: This expression combines the power operation from Expression 1 with the sine function from the same expression. The `(theta ^ log(theta / theta))` term is taken from Expression 1, and the `sin(C - sqrt(theta))` term is also from Expression 1. The `+ C` term is added at the end.

**(Assistant)**
Formatted
JSON
Summary

Here is the list of proposed expressions in JSON format:

```json
["log(exp(C - sqrt(theta ^ C)) + C)",
 "(sin(C - sqrt(theta)) * (theta ^ log(theta / theta)) + C)",
 "exp(C - sqrt(theta)) * sin(C - sqrt(theta))",
 "log(theta ^ log(theta / theta)) + C - sqrt(theta ^ C)",
 "(sin(C - sqrt(theta)) ^ (theta ^ log(theta / theta)) + C)"]
```

Figure 4: LLMCROSSOVER prompt with an example output. LLMMUTATION and LLMINIT follow the same structure but with slightly different wording and with one and no reference expressions, respectively. Variables within double braces are replaced with the instance specific arguments. These prompts are available in `prompts/*.txt` in the linked repository.

**(System)**
Domain
Specific
Header

You are a helpful assistant that hypothesizes about the underlying assumptions that generated a list of good and bad mathematical expressions in detailed ways. My ultimate goal is to discover what assumptions generated the observed good mathematical expressions and excludes the bad mathematical expressions. Focus more on the good expressions, their mathematical structure, and any relation to physical concepts. Note that capital C represents an arbitrary constant.

**(User)**
Concept
Abstraction
Prompt

Good Expression 1: **{{gexpr1}}**
Good Expression 2: **{{gexpr2}}**
Good Expression 3: **{{gexpr3}}**
Good Expression 4: **{{gexpr4}}**
Good Expression 5: **{{gexpr5}}**

Bad Expression 1: **{{bexpr1}}**
Bad Expression 2: **{{bexpr2}}**
Bad Expression 3: **{{bexpr3}}**
Bad Expression 4: **{{bexpr4}}**
Bad Expression 5: **{{bexpr5}}**

Propose {{N}} expressions that would be appropriate given the suggestions and expressions. Provide short commentary for each of your decisions. End with a JSON list that enumerates the proposed expressions following this format:

**(User)**
JSON
Formatting
Instructions

```json
["hyp1",
 "hyp2",
 ...
 "hyp{{N}}"
]
```

**(Assistant)**
Synthesized
Concepts
with
Explanation

**Hypothesis 1:** The good expressions involve complex exponential or trigonometric functions with nested power operations, potentially indicating a link to physical phenomena that exhibit exponential growth or oscillatory behavior.

**Hypothesis 2:** The presence of sine and cosine functions in the good expressions suggests a connection to periodic phenomena in real-world applications, such as wave functions or harmonic oscillations.

**Hypothesis 3:** The inclusion of arbitrary constants in the good expressions may represent parameters in physical models, implying that the expressions are modeling dynamic systems or relationships between variables.

**Hypothesis 4:** The good expressions involve repeated application of mathematical operations (e.g., exponentiation, multiplication, and addition) in a layered manner, indicating a depth of complexity that mirrors intricate physical systems or natural processes.

**Hypothesis 5:** The mathematical relationships within the good expressions exhibit a balance between simplicity and complexity, suggesting that they capture fundamental principles or underlying structures that govern physical phenomena.

**(Assistant)**
Formatted
JSON
Summary

Here is the list of proposed expressions in JSON format:

```json
["The good expressions involve complex exponential or trigonometric functions
with nested power operations, potentially indicating a link to physical
phenomena that exhibit exponential growth or oscillatory behavior.",
 "The presence of sine and cosine functions in the good expressions suggests a
connection to periodic phenomena in real-world applications, such as wave
functions or harmonic oscillations.",
 "The inclusion of arbitrary constants in the good expressions may represent
parameters in physical models, implying that the expressions are modeling
dynamic systems or relationships between variables.",
 "The good expressions involve repeated application of mathematical operations
(e.g., exponentiation, multiplication, and addition) in a layered manner,
indicating a depth of complexity that mirrors intricate physical systems or
natural processes.",
 "The mathematical relationships within the good expressions exhibit a balance
between simplicity and complexity, suggesting that they capture fundamental
principles or underlying structures that govern physical phenomena."]
```

Figure 5: LLM Concept Abstraction prompt with an example output. The LLM Concept Crossover function follows a similar structure, with a modified task description for crossover on concepts.

**(System)**
Header

You are a helpful assistant that merges and refines ideas about a set of hidden mathematical expression in new, interesting, and diverse ways. My ultimate goal is to discover the underlying properties of these hidden expressions. The resulting ideas should be a nontrivial conclusion given the previous ideas.

**(User)**
Concept
Abstraction
Prompt

Idea 1: **{{idea1}}**
Idea 2: **{{idea2}}**
Idea 3: **{{idea3}}**
Idea 4: **{{idea4}}**
Idea 5: **{{idea5}}**

Propose {{N}} hypotheses that would be appropriate given the ideas. Provide short commentary for each of your decisions. Do not talk about topics related to the simplicity or complexity of the expressions. I want ideas that are unique and interesting enough to amaze the world's best mathematicians. End with a JSON list that enumerates the proposed hypotheses following this format:

**(User)**
JSON
Formatting
Instructions

```json
["hyp1",
 "hyp2",
 ...
 "hyp{{N}}"
]
```

**(Assistant)**
Synthesized
Concepts
with
Explanation

**Hypothesis 1: Temperature Modulated Exponential Decay in Dynamical Systems.** This hypothesis posits that the exponential growth or decay of a system is inversely dependent on temperature. As temperature increases, the decay rate of the system's state energy function decreases, leading to a slower approach to equilibrium. Conversely, as temperature decreases, the system's decay rate increases, leading to a faster stabilization. This relationship suggests that temperature acts as a modulating factor in the dynamical system's behavior, potentially introducing non-linear effects.

**Hypothesis 2: Temperature-Driven Phase Transitions in Dynamical Systems.** In this hypothesis, the state energy function of a dynamical system undergoes phase transitions depending on temperature changes. These transitions can be modeled by exponential functions that describe the shift from one state to another, suggesting that critical points in the temperature spectrum trigger significant changes in the system's behavior. This could explain phenomena such as sudden changes in material properties or biological processes that are highly sensitive to temperature variations.

**Hypothesis 3: Entropy-Based Energy Functions in Temperature-Dependent Dynamical Systems.** Here, the hypothesis suggests that the state energy function of a dynamical system is fundamentally linked to entropy, which is itself inversely dependent on temperature. The dynamical system's behavior can be described through an entropy-based energy function that exhibits exponential growth or decay. This approach merges thermodynamics with dynamical systems theory, proposing that changes in entropy, driven by temperature variations, dictate the system's evolution over time.

Here is the list of proposed expressions in JSON format:

**(Assistant)**
Formatted
JSON
Summary

```json
[
  "Temperature Modulated Exponential Decay in Dynamical Systems",
  "Temperature-Driven Phase Transitions in Dynamical Systems",
  "Entropy-Based Energy Functions in Temperature-Dependent Dynamical Systems"
]
```

Figure 6: LLM Concept Evolution prompt with an example output. The LLM Concept Evolution prompt follows a similar structure to the concept abstraction and LLM operation prompts with slight modifications.

```
niterations=40,
ncyclesperiteration=550,
populations=15,
population_size=33,
maxsize=30,
binary_operators=["+", "*", "-", "/", "^"],
unary_operators=["exp","log","sqrt","sin","cos"],
weight_randomize=0.1,
nested_constraints={"sin": {"sin": 0, "cos": 0},
                    "cos": {"sin": 0, "cos": 0},
                    "exp": {"exp":0, "log": 0},
                    "log": {"exp": 0, "log": 0},
                    "sqrt": {"sqrt": 0}},
constraints={"sin": 10, "cos": 10,
             "exp": 20, "log": 20, "sqrt": 20,
             "pow": (-1, 20)},
```

Figure 7: The PySR hyperparameters used in all experiments. Whenever possible, we use the default PySR parameters.

### A.3.3 Hyperparameters

Figure 7 showcases the hyperparameters used for all our experiments. Wherever possible, we use the default PySR parameters. Additionally, LASR introduces three new hyperparameters: (1) % of LLM calls, (2) List of user hints, and (3) a dictionary of parameters pertaining to backend LLM communication. Following other methods in SRBench, we utilize only a subset of the necessary operators for solving the Feynman equations, excluding special operators like arcsin and arctan. These operators are seldom required, and removing them speeds up the search process. We generally set the number of iterations to 40. However, certain experiments may demand more or less iterations.

### A.4 Dataset Details

### A.4.1 Feynman Equations

**Dataset**: The Feynman Equation dataset is a widely adopted benchmark for scientific discovery [49]. The dataset consists of 100 physics equations extracted from the Feynman lectures on Physics. Each equation is in the form $y = f(x_1, x_2, \dots)$. The number of input variables ranges from two to ten, and the dataset provides 100,000 samples for each equation. We compare against publically available methods benchmarked on SRBench [26]. SRBench is a continuously updated benchmark which catalogs the performance of various methods on the Feynman dataset as well as other symbolic regression problems. Specifically, we compare against GPlearn, AFP, AFP-FE, DSR, uDSR, PySR, and the original AI Feynman algorithm [42, 47, 49, 28, 38]. Within this subset, notably, PySR represents an ablation of our model without the LLM genetic operations and the concept evolution (Section 3). We evaluate on a slightly noisy version of this dataset in order to simulate experimental errors common in scientific discovery domains. Specifically, we compare numbers against those reported in and reproduced by SRBench with a target noise of 0.001.

**Methodology**: For the Feynman dataset, we took the equations and the bounds at which each variable was sampled at and generated our dataset. Then, we added additional noise of 0.001 to our target variable, following the noise formula detailed in the Appendix A.4 of [26], as well as additional random noise variables with arbitrary names to force the model for proper feature selection. We then evaluate exact matches by looking at if the predicted equation symbolically simplifies into the ground truth equation. For the ablation graphs, we used the PySR hyperparameter "early_stop_condition" to check if there is a "solution" after $N$ iterations.

### A.4.2 Synthetic Dataset

For the synthetic dataset, we ran a script that generates uncommon mathematical hypotheses that satisfy our constraints at random. Then, we ran PySR for 400 iterations and found all the equations that PySR performed poorly in, i.e. MSE loss greater than 1, while having a complexity less than 20.

| Metric | PySR ($10^6$ Iterations, 10 hour timeout) | LaSR (40 iterations) |
|---|---|---|
| Exact Solve | $59 + 3/100$ | **72/100** |

Table 5: An asymmetric comparison of PySR and LASR on the Feynman equations dataset (A.5.1). We run PySR for 10 hours per equation (thresholded to $10^6$ iterations) and compare the exact solve rate with that of LASR run for 40 iterations. We find that PySR is able to discover three more equations, but LASR still substantially outperforms PySR.

For these 41 remaining equations, we then compared LASR and PySR after 20 iterations using the average of their test set $R^2$ for each hypothesis.

## A.5 Additional Experiments

This section highlights additional experiments on various benchmarks to further characterize LASR's performance.

### A.5.1 Asymmetric comparison with PySR

A common concern with evaluating genetic optimization algorithms w.r.t number of iterations is that the 'hard stop' after a certain number of iterations might yield populations that haven't fully converged to their optimal values.

To account for this discrepancy in performance, we conduct an asymmetric comparison with PySR on the Feynman Equations dataset. Specifically, we allow PySR to run uninterrupted for 10 hours per equation, with the maximum number of iterations set to $1 \times 10^6$. We compare the results with that of LASR run interrupted for 40 iterations per equation (hence the asymmetric comparison). The results are detailed in Table 5.

Overall, we find that, despite running PySR substantially longer than LASR, LASR still substantially outperforms PySR. This is because PySR, like other evolutionary algorithms, is susceptible to falling in local minima and converges to this local minima extremely fast. This indicates that supplementing 'local search' strategies with LLM guidance is useful for symbolic regression.

### A.5.2 Subset of equations discovered by LaSR on the Feynman Dataset

| Equation Number & Reference | Ground Truth Equation | Discovered Equation |
|---|---|---|
| Equation 2 (I.6.20) | $p(x) = \frac{1}{\sigma\sqrt{2\pi}} e^{-x^2/2\sigma^2}$, | $f = \left(\frac{0.46314177}{\sigma}\right)\left(0.6059228^{\left(\sqrt{\frac{E}{\sigma}}\right)^{3.994391}}\right) \times 0.86212635$ |
| Equation 7 (I.11.19) | $\mathbf{a} \cdot \mathbf{b} = a_x b_x + a_y b_y + a_z b_z$, | $A = (x_1 y_1) + ((x_2 y_2) + (x_3 y_3))$ |
| Equation 51 (I.50.26) | $x_{\text{out}}(t) = K(\cos\omega t + \epsilon\cos^2\omega t)$. | $x = \left(\left(\left(\cos(\omega t)\left(x_1(\alpha - 2.991099 \times 10^{-8})\right) - 1.37413245 \times 10^{-8}\right) + x_1\right)\cos(\omega t) + 1.575935\right) - 1.3385427 - 0.23739222$ |
| Equation 21 (I.18.4) | $\mathbf{R} = \frac{m_1\mathbf{r_1}+m_2\mathbf{r_2}}{m_1+m_2}$ | $r = \frac{(m_1 r_1)+(m_2 r_2)}{m_1+m_2}$ |
| Equation 63 (II.11.20) | $P = \frac{Np_d^2 E}{3kT}$. | $\text{Pol} = \left(\frac{\frac{N_\rho p_d}{E_f}}{3.0001059}\right)p_d$ |
| Equation 96 (III.15.14) | $m_{\text{eff}} = \frac{\hbar^2}{2Ab^2}$. | $m = h\left(\frac{h}{E_n \cdot 0.8882483 + 5.0833223 \times 10^{-5}} - 0.00011104094\right)\frac{1}{d^2} \cdot 0.011250258$ |
| Equation 57 (II.6.15b) | $E_\perp = \frac{p}{4\pi\epsilon_0}\frac{3\cos\theta\sin\theta}{r^3}$. | $E_f = \left(\frac{p_d}{r} \cdot \frac{\cos(\theta)\sin(\theta)}{2.037181} \cdot 0.39283985\right)\frac{1}{r^2\epsilon} \cdot \left(r^{0.00012676959} + 0.23802117\right)$ |

Table 6: A subset of equations discovered by LASR on the Feynman Equations dataset (over PYSR). The equations are presented in the form discovered by LASR, and usually reduce to the ground truth equations after some simplification steps. Note that there are minor discrepancies in the variable names between the ground truth equations found in the online Feynman Lectures (https://www.feynmanlectures.caltech.edu) and those in our Feynman Equations dataset.

| Ground Truth Equation | Discovered Equation |
|---|---|
| $\exp\left(\frac{C-\log(y_2)}{C-y_4} + \left(\sqrt{y_1+y_4} + \sqrt{y_2}\right)\right)$ | $\left(\frac{(y_4+C+y_1)}{C}\cdot(y_2+C)\right)^C + y_1$ |
| $\frac{(y_1+C)+\left(\sqrt{y_3}\cdot(y_1\cdot(C-y_3)\cdot y_3)\right)}{\cos(\cos(y_1))}$ | $y_3\cdot\frac{C-(y_1\cdot(y_3-(\cos((y_1+C)\cdot C)\cdot C)+C)))}{C} - C$ |
| $\sqrt{\exp(y_1)\cdot y_1}\cdot(\cos(y_1)+2y_1)$ | $\left(y_1^C+C\right)\cdot\left(\sqrt{\exp(y_1)}-C\right) +$ $\left(C - \left(\frac{\left(y_1^2\cdot\exp(\cos(y_1))\right)}{C} + y_4 - C\right)\right)$ |
| $\exp\left(\log(y_4+y_2\cdot y_1)\right)\cdot$ $\left(\cos(y_2)-\sqrt{y_1}+\frac{y_3}{C}-C\right)$ | $y_1\cdot\left((y_2\cdot\cos(y_2))-y_4-(y_3+1)\cdot y_2\right)-1$ |
| $(y_3+y_2)\cdot\left(\left((y_3+C)\cdot y_2+y_1+y_3\right)-C\right)$ | $((y_2+y_3)\cdot(y_3+(y_2\cdot(y_3+1))+1+y_1)-C+y_3)$ |
| $\left(\sqrt{y_2}+\exp\left(C+\sqrt{y_2}\right)\right)\cdot\left((y_4\cdot y_3)\cdot\log(y_3)\right)$ | $\frac{\left(\left(\sqrt{y_2^C}-C\right)\cdot(y_4\cdot C\cdot(y_3-C))-\sin\left(y_2^C+C\right)\right)\cdot\log(y_3)}{C} + C$ |
| $\frac{(C\cdot y_3)\cdot\exp(\sqrt{y_2}-y_1)}{C/(y_2+y_1)}$ | $\left(y_2\cdot\frac{C-\sqrt{y_2^C}}{\left(\sqrt{y_1^C}+C\right)^{y_1}}\right)\cdot y_3 - C$ |
| $\frac{(y_2+C)\cdot\sqrt{y_1}}{C}\cdot\sqrt{\frac{\exp(y_1)}{y_1}}$ | $\frac{C-y_2}{\sin\left(\frac{C}{y_1+C}\right)^C} - \cos(C-y_1)$ |
| $y_1\cdot\left(\frac{y_1^2}{\frac{C}{\cos(C-y_4-C)}}\right)/\exp(C\cdot y_5)$ | $((y_1-C)\cdot(\sin(y_4+C)\cdot y_1)\cdot(y_5+y_1-C))\cdot$ $C-C$ |
| $\exp(\log(y_3)\cdot\sqrt{y_2}) -$ $\exp\left(\sqrt{(y_3+y_4)}+(y_1\cdot y_2)\cdot C\right)$ | $\left(\frac{y_3\cdot(y_2/C)^C+C-y_4-y_2}{-y_2}\right)$ |

Table 7: Qualitative evaluation of LASR on the synthetic equations dataset (A.5.3). We attempt to recover the ground truth equation from a slightly noisy dataset generated from the ground truth equations. None of the algorithms we tested (including LASR) are able to recover the ground truth equations, underscoring the challenge of exact symbolic match on this dataset. A study on the $R^2$ performance is presented in Table 3.

.

### A.5.3 Qualitative comparison of Synthetic Dataset equations

In Table 3, we reported $R^2$ test set performance of LASR and PySR on the synthetic dataset. In this section, we qualitatively compare the equations discovered by LASR and PySR on the procedurally generated dataset of equations. None of the algorithms recover the ground truth form, but we find that LASR's equations fit much better to the ground truth data than PySR's equations. These equations are presented in Table 3.

### A.5.4 Stochasticity of LASR and PySR

A common concern with using LLM's for scientific discovery is that the equations could be obtained due to dataset memorization rather than reasoning and learning on the equations form. To explore this further, we run LASR with the same hyperparameters twice with different seeds. We expect LASR to behave like PySR (or any stochastic evolutionary algorithm) and produce two syntactically different programs that achieve a similar data fitness score. However, if the algorithm purely relies on training set memorization, we expect the model to find the same equation form in both experiments. We present results in Table 8.

Overall, both equations fit well to the underlying dataset and reduce to the ground truth. Yet, despite using the same hyperparameters, the functional forms exhibit sharp differences. This further reinforces our hypothesis that LASR's performance is not simply the result of regurgitated memorized responses.

| Discovered Equation (LaSR - Llama 3b 1%; same params) | Loss | Complexity |
|---|---|---|
| $h = \frac{2.8841133}{0.000219 - \left(r \cdot \left(\frac{-36.240696 - (\sin(r) \cdot 0.002057)}{P}\right) \cdot r\right)}$ | $2 \times 10^{-6}$ | 16 |
| $h = \frac{P}{r \cdot (r \cdot 17.252695 - 0.001495)} \cdot 1.372847$ | $2 \times 10^{-6}$ | 11 |

Table 8: Performance of LaSR on successive runs with the same hyperparameters on Equation 53 in the Feynman equations dataset (A.5.4). This equation defines the heat flow from a point source (an inverse square law with ground truth formulation: $h = \frac{P}{4\pi R^2}$). We evaluate LaSR by running it twice with identical hyperparameters but different random seeds to assess whether the model has memorized the equation's form. In both runs, LaSR consistently discovers high-performing solutions. Moreover, the resulting functional forms show significant variation between runs, supporting the hypothesis that LaSR's performance is not rooted in memorization.

## A.6   Using LaSR to find an LLM Scaling Law

So far, we have used LaSR to discover equations that are already well-established in prior work. In this section, we investigate LaSR's utility in making novel empirical discoveries. Specifically, we investigate whether LaSR can discover novel scaling laws for LLMs.

**Motivation:**   Traditionally, to identify an LLM scaling law, practitioners must first manually posit a skeleton equation with a fixed set of known variables and unknown free parameters, and then optimize the unknown parameters based on a dataset of model hyperparameters and resulting dataset fitness [23, 1, 5]. Instead of starting with a predefined equation, we use LaSR to discover the skeleton equation that best fits various subsets of the BigBench dataset. Removing the need to manually posit a skeleton equation simplifies the methodology for finding scaling laws in many ways. First, it removes human preconceptions about the expected relationships between hyperparameters. Second, it increases the number of variables and the type of variables human practitioners can jointly reason about. Finally, it enables positing equations of much higher complexity and variable interdependence than otherwise possible.

**Dataset:**   BigBench is a collaborative benchmark intended to probe large language models and extrapolate their future capabilities [16]. It contains 204 tasks drawing upon problems from linguistics, cognitive science, math, physics, biology, software development, etc. For each task, Bigbench measures the performance on many model families (OpenAI's GPT models, Google's dense transformer architectures, and Switch-style sparse transformers). Each BigBench task is evaluated on a preferred metric (chosen by dataset creators). In our experiments, we consider a subset of tasks where the preferred metric is 'multiple choice grade.' This subset contains the highest diversity of tasks and around 53,812 total data points.

**Methodology:**   We run LaSR with 3840 populations (parallelized across 16 cores) with each population evolving 200 candidates evaluated with the Zygote autodifferentiation backend. We ran LaSR overnight (for 7 hours). BigBench necessitates optimizing a matrix of parameters rather than singular scalar constants. This shifts the compute bottleneck from generating a pool of candidates (in which LaSR is slower than PySR) to evaluating a pool of candidates (which is equally slow for both algorithms). We also impose regularization constraints to accelerate the optimization procedure and avoid degenerate equations. Specifically, we expect a monotonically increasing performance trend with respect to the number of training steps.

**Results:**   LaSR discovers the following scaling law on the subset of BigBench:

$$\texttt{score} = \frac{-0.0248235}{\left(\frac{\texttt{train\_steps}}{116050.999}\right)^{\texttt{\#shots}}} + 0.360124 \tag{5}$$

Where `score` is the MCQ grade, `train_steps` is the number of training steps for the model, and `#shots` is the number of in-context examples provided during inference. LaSR maintains a Pareto frontier of the simplicity-accuracy tradeoff between various best performing programs. We present all the programs in the pareto frontier of this experiment in Table 9.

| Complexity | Equation |
|---|---|
| 2 | $A$ |
| 4 | $A - B$ |
| 6 | $A + \left(\frac{-C}{A}\right)$ |
| 8 | $A + \left(\frac{-D}{\texttt{training\_steps}^{\texttt{number\_of\_shots}}}\right)$ |
| 10 | $A + \left(\frac{-D}{\left(\frac{\texttt{training\_steps}}{B}\right)^{\texttt{number\_of\_shots}}}\right)$ |
| 13 | $\frac{A+E}{\texttt{training\_steps}+\frac{\cos(\texttt{training\_steps})}{\cos(\texttt{training\_steps})+F}}$ |
| 16 | $\frac{A+G}{\left(\frac{H}{I+\cos(\texttt{training\_steps})}\cdot J\right)+(\texttt{training\_steps}-K)}$ |
| 17 | $\frac{E+A}{\texttt{training\_steps}+\frac{\cos(\texttt{training\_steps})}{\cos(\texttt{training\_steps})+(F-L\cdot B)}}$ |
| 19 | $\frac{A+M}{\left(\frac{N+A}{O+\cos(\texttt{training\_steps})}\cdot P\right)+(\texttt{training\_steps}-Q)}$ |
| 21 | $\frac{A+M}{\left(\frac{N+\cos(A+\texttt{total\_params})}{O+\cos(\texttt{training\_steps})}\cdot P\right)+(\texttt{training\_steps}-Q)}$ |
| 25 | $\frac{A+G}{\left(\texttt{training\_steps}-K+\left(\frac{H+\cos(A+\texttt{non\_embedding\_params})}{R+\cos(\texttt{training\_steps})}\cdot J-S\right)\right)\cdot T}$ |
| 26 | $\frac{A+G}{\left(\texttt{training\_steps}-K+\left(\frac{H+\cos(A+\texttt{non\_embedding\_params})}{R+\cos(\texttt{training\_steps})}\cdot J-S\right)\right)\cdot T}$ |
| 28 | $\frac{A+G}{\left(\texttt{training\_steps}-K+\left(\frac{H+B+\cos(A+\texttt{total\_params})}{I+\cos(\texttt{training\_steps})}\cdot J-S\right)\right)\cdot T}$ |
| 30 | $\frac{A+G}{\left(\texttt{training\_steps}-K+\left(\frac{H+B+\cos(A+\texttt{total\_params}^{U})}{I+\cos(\texttt{training\_steps})}\cdot J-S\right)\right)\cdot T}$ |
| 32 | $\frac{(A+G)-V}{\left(\texttt{training\_steps}-K+\left(\frac{\cos(A+\texttt{total\_params}^{W})+H+B}{I+\cos(\texttt{training\_steps})}\cdot J-S\right)\right)\cdot T}$ |

Table 9: Pareto frontier of LASR for the BigBench experiment (Section A.6). Equation 4 is shown as the equation with complexity 10 in this table.

*Qualitative Evaluation*: Equation 4 has some interesting properties. First, it describes an empirical relationship between training hyperparameters (training steps) and inference hyperparameters (number of shots). It asserts that increasing the number of shots exponentially increases the model's performance for low-resource models while having diminishing gains as the number of training steps of the model increase. Second, Equation 4 only requires three free parameters (Chinchilla [23] required five free parameters). This might allow LASR 's equation to better generalize to different LLM operation regimes. However, the equation has some issues as well. If the number of shots is an even number, the exponentially increasing trend reflects over the value of A and turns into an exponentially decreasing trend. A manual inspection of the training data reveals the majority of samples had an odd number of shots (1, 3, ...) which causes the generalization error.

*Quantitative Evaluation*: We compare Equation 4, Chinchilla [23], Modified Chinchilla, and a single free parameter equation on their ability to explain the score of a held out validation set of 10,763 data points. We fit the free parameters of each equation to the training set data and measure the MSE loss between the actual grade and the predicted grade on the validation set. The results are presented in Table 4. Overall, we find that the Equation 4's performance, as well as modified Chinchilla's performance, is competitive with that of Chinchilla's in predicting the MCQ grade.

We hope our preliminary results provide further insight into the practical value of LASR in addressing real-world challenges, including those in machine learning

### A.7 Metrics for Cascading experiment

For the cascading experiment, we aim to evaluate the progression of different configuration towards solving equations. The quantitative metric used in the SRBench comparison experiment, Exact Solve, does not allow for such fine-grained analysis. Therefore, we categorize the synthesized equations into four buckets: Exact Solve, Almost Solve, Close, and Not Close. Exact Solve is quantitatively evaluated using a symbolic match. An equation is tagged as 'Almost Solve' if the dataset loss is small, but the generated equation has an extra term or lacks one term. A Close equation captures the

general structure of the solution (such as a square root nested in an exponential) but not more than that, and Not Close includes all equations that are far from the solution.

## A.8 Further Qualitative Analysis

LASR generates two artifacts: the best fit program, and the library of natural language concept that helped find that program. These artifacts provide a unique window into the inner workings of LASR. This section goes over a qualitative study of how LASR and PySR go about discovering Coulomb's law $F = \frac{q_1 q_2}{4\pi r^2 \epsilon}$ from data. Both methods are able to find an answer to this equation. However, their approach to finding the best fit equation as well as the form of the equation they discover differs significantly.

**Setup**: Coulomb's law is equation #10 in the Feynman equation dataset. It describes how the force between two point charges changes with respect to the distance between the charges, the magnitudes of the charges, and the permittivity of free space constant. The corresponding data for this equation has a target noise of $0.001$ to simulate experimental errors.

By analyzing the form of the equation and relationships between variables in Coulomb's law, we can uncover several interesting properties: First, observe that this is an inverse square law (The force $F$ varies inversely with the square of the distance $r$ between the charged particles). Second, notice that the $F$ is directly proportional to the magnitude of the charges $q_1$ and $q_2$. Third, observe that the resultant force is symmetric with respect to the magnitude of the charged particles (i.e.: The magnitude of the $F$ doesn't change if the magnitude of the charged particles is swapped).

**PySR Solution**: PySR finds the following solution to this equation:

$$F = \frac{\left(\left(\left(\left(\left(\left(\frac{q_2 \cdot 3.382}{r}\right) - \left(\frac{\sin\left(\frac{0.017}{\exp(B)}\right)}{\exp(C)}\right)\right)/0.712\right) \cdot q_1\right) \cdot 0.087\right)/\epsilon\right) \cdot 0.191\right)}{r}$$

This equation has a complexity of 26 and achieves a loss of $2.191505 \times 10^{-12}$ on the dataset. Obtaining a simplification of this solution is rather painstaking.

**LASR's Solution**: LASR finds the following solution to this equation. We also present three steps of simplification:

$$F = \frac{q_1}{\left(\frac{r}{q_2}\right)\left(r + \frac{1.9181636 \times 10^{-5}}{q_2}\right)\epsilon} \cdot 0.07957782$$

$$= \frac{q_1}{\left(\frac{r}{q_2}\right)\left(r + \frac{1.9181636 \times 10^{-5}}{q_2}\right)\epsilon} \cdot \frac{1}{4\pi} \qquad \text{(Substitute constant)}$$

$$= \frac{q_1 q_2}{r\left(r + \frac{1.9181636 \times 10^{-5}}{q_2}\right)\epsilon} \cdot \frac{1}{4\pi} \qquad \text{(Simplify denominator)}$$

$$\approx \frac{q_1 q_2}{r\left(r\right)\epsilon} \cdot \frac{1}{4\pi} \qquad \text{(Negligible. } \frac{1.9181636 \times 10^{-5}}{q_2} \approx 0)$$

This equation has a complexity of 15 and achieves a much lower loss of $4.6709058 \times 10^{-14}$ on the accompanying dataset. We can see with just three steps of simplification how this equation might be reduced to the ground truth.

Let's examine some essential concepts from various iterations in the search process. Keep in mind that an LLM operates on *tokens* in each concept. Consequently, even small relevant substrings can positively influence future LLM inference calls, despite full concepts appearing verbose to humans.

1. **Iteration 2** *The good mathematical expressions exhibit a clear and coherent relationship between the variables involved, with a focus on **power functions and trigonometric functions** that can be easily related to physical concepts.*

2. **Iteration 6** *The good mathematical expressions exhibit a certain level of **symmetry or regularity in their form**, possibly reflecting underlying patterns or relationships between the variables and constants.*

3. **Iteration 24**: *The good mathematical expressions have a clear and consistent structure involving the variables q1, q2, epsilon, C, and r, with a specific pattern of **division and multiplication**.*

