# OpenReview forum: "Symbolic Regression with a Learned Concept Library"
_NeurIPS.cc/2024/Conference — NeurIPS 2024 poster_

### Official Review · Reviewer_CCz9 · 2024-07-08

**Soundness:** 4
**Presentation:** 4
**Contribution:** 4
**Rating:** 8
**Confidence:** 4

**Summary:**

This paper proposes a way to incorporate LLM prompting to improve symbolic regression. They uses PySR, a standard symbolic regression library, as their base SR algorithm. Then they add LLM prompts to different SR algorithm steps: population mutation, crossover, and initialization. They replace the PySR implementation with LLM prompted implementations 1% of the time. The LLM prompts are based on identifying high level concepts and prompting the LLM to do its operation using the concept as a suggestion to follow. High level concepts are tracked using an abstraction prompt given high performing expressions, and are evolved using LLM prompting as well. The authors show that augmenting PySR with LLM concept-based prompting solves around 7/100 addition tasks from the AI Feynman SR benchmark, and show that PySR + LLM (LaSR) performs better on a synthetic task designed to test for data leakage.

**Strengths:**

- Well written, great figures
- The framework for integrating LLM-based SR into PySR is well-designed, and could in principle work for other prompting approaches besides the concept-based SR. The LLM-mutation, crossover, and initialization steps could be replaced with other LLM based techniques. Cool!
- Using LLM's to learn concepts to guide SR is a well-motivated choice of prompting technique, given the importance of high level concepts for human equation discovery
- Based on the literature review comparing LaSR to two other LLM SR tools, LaSR seems like an original contribution.
- Given existing literature on program synthesis with library learning, LaSR is a great approach that bridges the gap a bit between SR and program synthesis, as done with modern tools.
- LaSR is also a good contribution to the growing body of work on library learning and its benefits for search. It is very similar to LiLO, which is built off DreamCoder, but applied to symbolic regression.
- The authors have a strong analysis of LLM incorporation based on (1) algorithmic cost in terms of millions of tokens per iteration, and (2) comparing GPT 3.5 with open source llama 8b on the results.

**Weaknesses:**

- It's not clear how valuable the concept abstraction approach is compared to some "baseline" of simple LLM prompting. For example, one baseline could just use a single concept "This expression is a good formula for symbolic regression" or something like that, and see how it compares. This could perhaps be a direction for future work: try a bunch of simple prompting strategies for combining LLM's with PySR, and report how well each of them work.
- I'm not sure what to make of the synthetic dataset. In particular, PySR works so well on AI Feynman alone, but works very poorly without the LLM addition on this synthetic dataset. Why does LaSR work so much better than PySR here, but not help as much on AI Feynman? One hypothesis is that AI Feynman has a lot of easy tasks, and the synthetic benchmark only has tasks right on the edge of what PySR can discover, which LLM incorporation helps push over the edge. Another pessimistic take is that the synthetic benchmark is designed with LaSR in mind. While this still eliminates worries on data leakage, an explanation here would help understand better how LaSR is being helpful.

I'd like to emphasize that including answers to these questions (both of which could suggest negative results) in the paper would not decrease my review score.

**Questions:**

See weaknesses section.

Some other suggestions:
- fix backwards quotations marks

---

> ### Author Rebuttal · Authors · 2024-08-07
>
> Thank you for taking the time to review our work! We address your questions inline:
>
> > **It's not clear how valuable the concept abstraction approach is compared to some "baseline" of simple LLM prompting. For example, one baseline could just use a single concept "This expression is a good formula for symbolic regression" or something like that, and see how it compares. This could perhaps be a direction for future work: try a bunch of simple prompting strategies for combining LLM's with PySR, and report how well each of them work.**
>
> Good suggestion! This strategy – fixing a static (but highly domain specific) prompt – usually causes the LLM to regurgitate low performing programs and requires a bigger, slower model to work properly. One of our ablations is very close to the suggested experiment: in Figure 3 Left, we compare against LaSR with no concept evolution (in this case, “the fixed prompt” is just the system prompt) . Nevertheless, LaSR paves the way to try out exciting techniques from the prompt programming `[1]` / prompt tuning community for solving scientific discovery problems.
>
> **I'm not sure what to make of the synthetic dataset. In particular, PySR works so well on AI Feynman alone, but works very poorly without the LLM addition on this synthetic dataset. Why does LaSR work so much better than PySR here, but not help as much on AI Feynman? One hypothesis is that AI Feynman has a lot of easy tasks, and the synthetic benchmark only has tasks right on the edge of what PySR can discover, which LLM incorporation helps push over the edge.**
>
> Thank you for pointing this out. Randomly generated synthetic equations wouldn’t be informative about the capabilities of LaSR or PySR. To properly verify the utility of language guidance, we generated the synthetic dataset equations to be within the upper limit of PySR’s performance. To construct this dataset, we first generated equations which, anecdotally, contain many characteristics that PySR struggles with. Then, we verified PySR’s performance on this dataset for 400 iterations to ensure there weren’t trivial bugs in the equations (such as the equation evaluating to zero). This left us with 41 equations, for which we report LaSR’s performance with the same hyperparameters.
>
> **Another pessimistic take is that the synthetic benchmark is designed with LaSR in mind. While this still eliminates worries on data leakage, an explanation here would help understand better how LaSR is being helpful.**
>
> Thank you for pointing this out. As mentioned before, randomly generated synthetic equations wouldn’t be informative about the capabilities of LaSR or PySR. We deliberately want to pick equations that would be hard for PySR to properly verify the utility of language guidance. We’ve edited the synthetic dataset section to clarify the data construction process, which explains PySR’s poor performance.
>
> **Discovering what is known…**
>
> Our experiments on synthetic data suggest that we can use LaSR to make discoveries even without significant prior domain knowledge. In addition, since the submission, we have been actively exploring ways to use LaSR in novel discovery tasks. For example, we currently have exciting preliminary results on automatically discovering novel LLM scaling laws using LaSR. Such real-world case studies require deep collaboration with domain experts and are hence outside the scope of the present foundational effort. However, we anticipate many such results in the future.
>
> **Lack of Running Time**
>
> Our goal is to present experimental results that can be replicated reliably by the broader community. As such, we root our experiments in the number of iterations instead of wall time.
>
> This is because the wall time is heavily dependent on the LLM backend, the number of tokens each query uses, and the network quality. For instance, for `gpt3.5` we cannot reliably measure the runtime as the backend is slower during periods of high traffic. Even for local models, vLLM's query scheduling is non-deterministic so the wall time will be substantially different across experiments.
>
> Regardless, we were measuring around one second per query using vLLM. Since we make 60k calls per iteration (Appendix A.3.1) the amount of time for p=1% would be around 600 seconds / 10 minutes per iteration. PySR roughly takes 30 seconds per iteration. We expect performance to improve as we increase CPU/GPU parallelism, and as the hardware to run LLM inference improves.
>
> `[1]`: https://github.com/stanfordnlp/dspy

---

> > ### Comment · Reviewer_CCz9 · 2024-08-07
> >
> > I think your comment on runtime and discovering what is known belong in the response to a different review than mine.
> >
> > Thanks for your response. I will keep my score.

---

> > > ### Author Response · Authors · 2024-08-07
> > >
> > > Thank you for pointing this out. Apologies for any confusion this might have caused!

---

### Official Review · Reviewer_mCEz · 2024-07-10

**Soundness:** 3
**Presentation:** 3
**Contribution:** 3
**Rating:** 7
**Confidence:** 4

**Summary:**

This paper introduces a method that learns a library of concepts (natural language description of helpful and unhelpful concepts) as a means of guiding genetic search for symbolic regression. The core idea is that such concepts can be used to bias genetic operations through an LLM.

The method was evaluated on the 100 Feynman equations and on a synthetic dataset. The paper also includes ablation studies on the various components of the system.

**Strengths:**

The paper presents a creative form of using LLM to speed up the search of genetic algorithms for symbolic regression. Instead of simply storing a library of programs, as GA algorithms do, the algorithm also stores a library of natural language concepts. Such concepts can be seen as abstractions of the population of programs encountered in search. Given in natural language, the abstractions can be used to drive the search as an LLM can be used to generate programs based on the description. Such a creative approach!

The idea presented in this paper is general and can be more broadly applied to other problems. I can already see how I could use a similar approach in my own research!

Another strength of the approach is the author's care with data contamination. Initially, I was skeptical of the approach as it uses LLMs to solve problems whose solutions are available online. The authors then explain that the way the LLM is used is unlikely to allow it to simply retrieve the solution from its training data. The explanation makes perfect sense since the LLM is used to extract concepts from programs the GA generates, and they can't encode the solution available online. In addition to this explanation, the authors also included an experiment on synthetic data showing the advantages of the learned concepts over the search alone. Nicely done!

I also enjoyed the fact that the system is built on top of PySR, which is a very efficient system for symbolic regression. This eliminates the possibility that all the gains the LLM provides could be easily washed with clever engineering. The current results already show that clever engineering alone is outperformed by the system.

**Weaknesses:**

The paper also has a few weaknesses.

**Claims that need to be fixed**

Some of the claims in the paper are a bit strong and I suggest toning them down. While the leakage explanation the authors provided is reasonable, I would be careful in claiming state-of-the-art performance. When writing "LASR achieves a higher exact solve rate than all other baselines," it is worth mentioning the possibility of leakage.

Another claim that seems to be incorrect is the following: "LaSR's increasing the backbone model size and the mixture probability significantly enhances." I think the authors meant to write "substantially" and not "significantly" as there is no statistical test involved. I would also explain why these results are substantially better as the number of problems solved isn't much larger. The explanation I gave to myself is that solving each of these equations is very difficult, so solving one new equation is already quite an achievement.

Another claim that needs to be adjusted: "demonstrating that LaSR's performance gains are not rooted in memorized responses." The experiments on the synthetic dataset do not demonstrate this. The experiment with the synthetic dataset is almost independent of the experiment with the 100 Feynman equations. What the experiment demonstrates is that LaSR can outperform PySR even when data leakage is not possible.

**Discovering what is known**

Perhaps an unfair criticism of the paper is that the method it introduces is used to discover things we already know. I understand that the bar would be way too high and it would be unhealthy to the research area if we required the discovery of new things with the presentation of novel approaches to scientific discovery. So I do not make this criticism as a means of arguing for rejecting the paper (I think the paper should be accepted), but more as a reflection of what the community has been pursuing. The hope is that systems such as PySR and LaSR will eventually be used to make actual discoveries.

**Lack of Running Time**

I missed the running time of LaSR and PySR in the paper. How do they compare with LaSR making calls to an LLM?

**Questions:**

1. In line 83, shouldn't it be $P(C)$ instead of $P_C$?

2. In the mutation operator, it is stated that for each deleted subtree, one adds a single leaf node (lines 111-112). Is this correct? Why not add an entire new subtree?

3. In the for-loop in line 4 for Algorithm 1, shouldn't it be $I$ instead of $N$?

4. Why not provide a figure in the appendix to describe Concept Evolution (similar to figures 4 and 5)?

5. How does that cascade work for PySR? (the one that didn't work)

6. Why use MSE and not the exact correct metric in Section 4.3?

7. Why is the difference between PySR and LaSR so large in Table 3, but somewhat small in Table 1?

7. Weren't the sentences in Section 4.4 swapped? To me, the first sentence looks more refined than the second. This is because it mentions "waveforms or periodic phenomena," while the second only talks about "scientific phenomena." Perhaps the argument in that section needs some rethinking?

8. In the prompt shown in Figure 5, how much is the excerpt "relation to physical concepts" contributing to the quality of the concepts?

**Limitations:**

The paper lists all limitations I could think of, either in the last section "limitations" or throughout the paper.

---

> ### Author Rebuttal · Authors · 2024-08-07
>
> Thank you for taking the time to review our work! We address all questions inline:
>
> > **Claims that need to be fixed.**
>
> We would be happy to adjust these claims along the lines you suggest. Thank you for pointing this out!
>
> > **In line 83, shouldn't it be P(C) instead of P_C?**
>
> Good catch! Fixed.
>
> > **In the mutation operator, it is stated that for each deleted subtree, one adds a single leaf node (lines 111-112).**
>
> This isn’t correct. We’ve updated the explanation for `delete_subtree` on line 111. In PySR’s implementation, the delete_subtree function randomly selects a single node from the subtree, deletes that node, and replaces the deleted node with one of its children (selected randomly).
>
> >  **In the for-loop in line 4 for Algorithm 1, shouldn't it be I instead of N?**
>
> Yes, it should be. Fixed in the latest manuscript.
>
> > **Why not provide a figure in the appendix to describe Concept Evolution (similar to figures 4 and 5)?**
>
> Thank you for the suggestion! We added a figure for concept evolution in the appendix. The prompt should be available in the provided repository.
>
> > **How does that cascade work for PySR? (the one that didn't work)**
>
> The PySR cascade operates in a similar fashion to the LaSR cascade. We run PySR for 40 iterations, prune equations that are below a threshold solution, and repeat the process the same number of times as we do for LaSR. Pruning and rerunning PySR should have no effect on the performance but we do this anyway for completeness.
>
> >  **Why use MSE and not the exact correct metric in Section 4.3?**
>
> Due to rounding errors of the data, waiting for a MSE error of exactly zero (an exact match) is not ideal.
>
> > **Why is the difference between PySR and LaSR so large in Table 3?**
>
> Thank you for pointing this out. Randomly generated synthetic equations wouldn’t be informative about the capabilities of LaSR or PySR. To properly verify the utility of language guidance, we generated the synthetic dataset equations to be within the upper limit of PySR’s performance. To construct this dataset, we first generated equations which, anecdotally, contain many characteristics that PySR struggles with. Then, we verified PySR’s performance on this dataset for 400 iterations to ensure there weren’t trivial bugs in the equations (such as the equation evaluating to zero). This left us with 41 equations, for which we report LaSR’s performance with the same hyperparameters. We’ve edited the synthetic dataset section to clarify the data construction process, which explains PySR’s poor performance.
>
> > **Weren't the sentences in Section 4.4 swapped? To me, the first sentence looks more refined than the second. This is because it mentions "waveforms or periodic phenomena," while the second only talks about "scientific phenomena."**
>
> The second sentence is, in fact, refining upon the first sentence in some ways. For instance, the second sentence mentions more mathematical operations (multiplications, division, and trigonometric operations) than the first sentence.
>
> This is a common challenge with using LLMs: some parts of the generated concept undergo more refinement than other parts do. And it’s hard to quantitatively contrast two natural language concepts. Nevertheless, this additional generation flexibility is advantageous in many situations because it enables the LLM to abstract similar parts of the sentence between iterations.
>
> > **In the prompt shown in Figure 5, how much is the excerpt "relation to physical concepts" contributing to the quality of the concepts?**
>
> We didn’t find any performance difference including/excluding that phrase from the prompt. In our experience, the instruction fine tuned models are most sensitive to the initial instructions, the formatting instructions, and the bullet point list of concepts/equations.

---

> > ### Author Response · Authors · 2024-08-07
> > **Slight corrections.**
> >
> > Due to a formatting error, it seems part of our response to your questions ended up in our response to Reviewer CCz9. We copy the relevant portions verbatim in this comment:
> >
> > > **Discovering what is known…**
> >
> > Our experiments on synthetic data suggest that we can use LaSR to make discoveries even without significant prior domain knowledge. In addition, since the submission, we have been actively exploring ways to use LaSR in novel discovery tasks. For example, we currently have exciting preliminary results on automatically discovering novel LLM scaling laws using LaSR. Such real-world case studies require deep collaboration with domain experts and are hence outside the scope of the present foundational effort. However, we anticipate many such results in the future.
> >
> > > **Lack of Running Time**
> >
> > Our goal is to present experimental results that can be replicated reliably by the broader community. As such, we root our experiments in the number of iterations instead of wall time.
> >
> > This is because the wall time is heavily dependent on the LLM backend, the number of tokens each query uses, and the network quality. For instance, for `gpt3.5` we cannot reliably measure the runtime as the backend is slower during periods of high traffic. Even for local models, vLLM's query scheduling is non-deterministic so the wall time will be substantially different across experiments.
> >
> > Regardless, we were measuring around one second per query using vLLM. Since we make 60k calls per iteration (Appendix A.3.1) the amount of time for p=1% would be around 600 seconds / 10 minutes per iteration. PySR roughly takes 30 seconds per iteration. We expect performance to improve as we increase CPU/GPU parallelism, and as the hardware to run LLM inference improves.
> >
> > `[1]`: https://github.com/stanfordnlp/dspy

---

> > > ### Comment · Reviewer_mCEz · 2024-08-14
> > >
> > > Thank you for answering all my questions and fixing all the small issues.

---

### Official Review · Reviewer_2Egm · 2024-07-12

**Soundness:** 3
**Presentation:** 4
**Contribution:** 3
**Rating:** 6
**Confidence:** 4

**Summary:**

This work focus on symbolic regression. They enhaned the traditional method like genetic algorithms by inducting a library of abstract textual concepts. The algorithm, called LASR, uses zero-shot queries to a large language model to discover and evolve concepts occurring in known high-performing model to discover and evolve concepts occurring in known high-performing hypotheses. The algorithm can be seen as a kind of hybird of evolutionary algorithm and LLMs. Through experiments, LASR substantially outperforms a variety of state-of-the-art SR approaches.

**Strengths:**

1. This work proposed to introduce a concept library in symbolic regression, which is really similar to how human works, so the idea make sense. For the introduction of the library, this work also leverage abstrction or understanding ability of LLMs to design three phrases process, concept evolution, hypothesis evolution and concept abstraction. The design mix the strenghes of evolutionary algorithms and LLMs.
2. The experimental results are good comparing to those other baselines of learning-based or evolutionary-based.

**Weaknesses:**

1. Introdution of LLMs would inevitably raise the cost for the task comparing those traditional algorithms.
2. (this could be a question) The introdution of concept library seems to not making sense in all cases, imaging that we are facing some totally unknown black-box environment, the backbone function could be something random or out of the knowledge we have. In this setting, the traditional evolutionary algorithms might perform better because they do not have this kind of knowledge as the constrant.

**Questions:**

1. What's the operation of LLMINIT? If we do not have any prior knowledge, then what will we prompt LLMs to do the initialization? Is it the same when we nothing more to know in advance.
2. The initilization, mutation and crossover operation in LASR hybrid traditional evolutionary algorithms and LLMs, so what's the intuition to do this? Because there are several works that directly replace genetic operation with LLMs operation, such as (https://arxiv.org/abs/2401.02051). I am curious about have you ever tried to use merely LLMs for the whole process, that would be an interesting topic.
3. Is LASR capable to resolve high-dimentional symbolic regression tasks? Because current research about symbolic regression struggle to due with high-dimensional settings.
4. Have you ever tried to compared the work KAN (https://arxiv.org/abs/2404.19756), which can also be used in symbolic regression and report a great results.

**Limitations:**

See weakness and question parts.

---

> ### Author Rebuttal · Authors · 2024-08-07
>
> Thank you for taking the time to review our work! We address questions inline:
>
> >  **Introduction (...) algorithms.**
>
> We agree that LLMs might require additional compute, which can raise the cost. However, this is a worthwhile tradeoff for scientific discovery for a couple of reasons:
>
> - The expected reward of finding even one additional equation is extremely high. As LaSR consistently outperforms PySR, the additional cost of running LaSR is worth the tradeoff.
> - We are optimistic that the cost of running LaSR will further decrease – not increase – as local language models become faster, smaller, more accurate, and cheaper in the near future `[1]`. We are even able to run LaSR on a laptop with the latest quantized models!
> - LaSR usually considers a *larger* space of equations than a traditional symbolic method in the same number of iterations. Furthermore, LaSR produces two artifacts: an equation of best fit as well as a library of concepts the method finds relevant to discovering that equation. These provide invaluable context information to a scientist.
>
> > **(...) black-box environment, the backbone (...) traditional evolutionary algorithms might perform better because they do not have this kind of knowledge as the constraint.**
>
> This is exactly the setting of our Synthetic dataset experiment. We procedurally generate novel synthetic equations, and their respective data points, that lie outside the distribution of known equations, and find that LaSR is able to solve more equations than PySR – an ablation of LaSR without language guidance.
>
> The reason LaSR works in such scenarios is because LaSR is a *neurosymbolic* approach which uses a classical evolutionary search as the backbone and then uses the LLM to direct this search. In the synthetic domain, the novel synthetic blackbox equations generated through the evolutionary search still exhibit abstract mathematical trends, and LLMs can discover these trends, and then use these trends to guide further search.
>
> > **What's the operation of LLMINIT? If we do not have any prior knowledge, then what will we prompt LLMs to do the initialization? Is it the same when we nothing more to know in advance.**
>
> Great question! In the absence of prior knowledge / hints, we fall back to the symbolic initialization function. The full prompt is available in the linked repository.
>
> > **Have you ever tried to use merely LLMs for the whole process, that would be an interesting topic.**
>
> Thank you for the reference! Using an LLM for the whole process would not be ideal in cases where we do not have much prior concrete knowledge about the problem. For instance, in a completely black box setting, in LaSR, the LLM helps guide the evolutionary search using abstract concepts extracted by the classical evolutionary operations while exploring the “local” search space.
>
> Nevertheless, LaSR paves the way to try out exciting techniques from the prompt programming `[2]` / prompt tuning community for solving scientific discovery problems.
>
> >  **high-dimensional symbolic regression tasks**
>
> For high dimensionality problems, the SR community relies on the model distillation paradigm `[3]`. Here, we first fit a sparse GNN (or any large parametrized network with a sparsity constraint) to the data and then distill each network subcomponent into an independent equation. As LaSR is a drop-in replacement for PySR, a practitioner can use LaSR with this methodology to learn equations in high-dimensional tasks with almost no code changes. We leave this for future work.
>
> > **Comparison with KAN**
>
> Great suggestion! KAN’s `[4]` take a different, orthogonal approach as compared to our work for symbolic regression. Symbolic regression with KAN’s relies on a two-step model distillation strategy similar to `[3]`. First, they fit a sparse KAN to the dataset and then they use a “search” component to find the closest fit symbolic function for each learned activation (the search component here is iterative greedy refinement).
>
> Consecutively, the closest direct comparison of LaSR with KAN’s would be to measure the efficacy of greedy iterative refinement compared to LaSR for model distillation. This is an exciting direction for future work!
>
> `[1]`: https://arxiv.org/abs/2407.21783
>
> `[2]`: https://github.com/stanfordnlp/dspy
>
> `[3]`: https://arxiv.org/abs/2006.11287
>
> `[4]`: https://arxiv.org/pdf/2404.19756

---

> > ### Comment · Reviewer_2Egm · 2024-08-09
> > **Response to Authors**
> >
> > Thanks for your response! All of my concerns have been addressed. Good work and I will maintain my score.

---

### Official Review · Reviewer_1Xi9 · 2024-07-13

**Soundness:** 2
**Presentation:** 3
**Contribution:** 2
**Rating:** 4
**Confidence:** 5

**Summary:**

This paper introduces LASR, a symbolic regression framework that enhances PySR by incorporating Large Language Models (LLMs) to discover and evolve "concepts" from high-performing equations. These concepts are then used to guide the search process. LASR is evaluated on the Feynman equations dataset and a set of synthetic tasks, showing improved performance over existing symbolic regression baselines.

**Strengths:**

The idea of integrating LLMs into symbolic regression to learn and use abstract concepts in terms of natural language is interesting. The methodology is well-structured  with clear explanations of the algorithm components.

**Weaknesses:**

My main concerns are regarding the evaluation. Experimental design and analysis are insufficient to convincingly demonstrate the method's advantages over existing approaches; specifically: There are serious concerns about potential data leakage and unfair advantage when using LLMs on well-known, simple physics equations that may be part of LLM training data. While there is a section on the data leakage validation, its limited evaluation scope and lack of comprehensive analysis on more complex and real-world datasets, makes it difficult to assess the true capabilities and generalizability of the method.

**Questions:**

* Could you provide the 7 additional equations that LASR solves beyond PySR as well as the forms found by LASR? Are these 7 equations consistent across runs? What have been specific contributions of the LLM for them?

* How can you ensure that the LLM is not implicitly using its prior knowledge to generate simple Feynman physics equations? The ablation study shows that variable names and hints improve performance, which raises concerns. For instance, variable names like "g" for gravity or "Kb" for the Boltzmann constant could trigger the LLM to suggest the well-known relevant equations.

* Have you conducted ablation studies that remove all physics-related terms, variable names, and any mention of physics or internal knowledge from the prompts? This would help (to some extent) isolate the LLM's ability to learn purely from data patterns.

* For equations solved by both LASR and PySR, could you provide the number of iterations required to obtain the correct form using each method?

* Could you clarify what constitutes an "iteration" in your experiments? There are discrepancies between Figure 1 (10^6 iterations), the main experiments (40 iterations), and the synthetic data experiments (400 iterations). Could you explain these differences and provide results for more iterations (maybe around 10^6 iterations), particularly for the synthetic dataset? Additionally, could you clarify how the concept library operations frequency (every 400 iteration based on Figure 2), and LLM usage frequency (p=1%) translate to these small iteration counts? For example, for 40 iterations, how many time LLM operations and concept library operations will be used?

* For the data leakage validation, why were custom synthetic equations used instead of existing datasets like black-box datasets in SRBench? This would provide a more standardized comparison and better demonstrate LASR's capabilities on real-world problems.

* Could you provide a computation time comparison between LASR and PySR for experiments with more iterations, to assess the computational trade-offs of using LLMs?

* Could you provide examples of "hints" that are provided in experiments of Feynman equations?

* Based on Table 2, increasing the value of p seems to improve performance. Have you considered analyzing higher values of p, especially for GPT-3.5? Could you explain why higher values were not explored?

**Limitations:**

See above.

---

> ### Author Rebuttal · Authors · 2024-08-07
>
> Thank you for taking the time to review our work! We address questions inline:
>
> > **Could you provide the 7 additional equations that LASR solves ...?**
>
> We’ve added a table containing these equations and the ground truth equation to the global PDF for figures.
>
> > **Are these 7 equations consistent across runs?**
>
> Yes, LaSR discovers these equations consistently across runs.
>
> >  **What have been specific contributions of the LLM for them?**
>
> Due to the nature of evolutionary algorithms, it is extremely challenging to trace how each operation (whether it be an LLM-based operation or a symbolic operation) interacts with other operations to evolve a candidate pool. Intuitively, the LLM operations help introduce new candidates in the population that respect the natural language hypotheses (whether learned or user provided). LLM-biased candidates that are a good fit are retained by the evolutionary search, while bad candidates are discarded.
>
> We refer the reviewer to the qualitative study in the Appendix of a Feynman equation task, where we compare the equation LaSR finds with the equation PySR – an ablation of LaSR without the LLM operations – finds. Notably, we find that LaSR’s equation is more interpretable, yet is very far from the common formatting of the equation (available on the internet and, possibly, in the training dataset of the LLM).
>
> > **How can you ensure that the LLM is not implicitly using its prior knowledge ...**.
>
> We refer the reviewer to Figure 3 (Left) where the ablation of LaSR *without the variable names and the hints* still outperforms PySR.
>
> We find that more language guidance consistently improves LaSR’s performance (by adding variable names, by adding hints, by using a larger model, or by calling the LLM more frequently).
>
> Furthermore, In Appendix A.4.1, we analyze the equations discovered by PySR and LaSR from data corresponding to Coulomb's law. If LaSR’s performance was rooted in regurgitating memorized data, LaSR’s equation would correspond to the common formatting of Coulomb's law (F= (1/(4 * \pi * epsilon)) q_1.q_2/r^2). Instead, we observe that LaSR’s discovered equation displays a very different format, and needs at least five simplification steps to transform to the commonly seen equation.
>
> > **Have you conducted ablation studies...**
>
> Yes. Figure 3 (Left) depicts an ablation of LaSR without any physics knowledge (variable names or hints). This ablation still outperforms PySR. Also, one of our prompts contains the term “physical concepts.” We’ve verified that inclusion/exclusion of this phrase has no impact on LaSR’s performance.
>
> > **For equations solved by both LASR and PySR, could you provide the number of iterations required to obtain the correct form using each method?**
>
> Figure 3 tracks the number of equations solved after each iteration for PySR, LaSR, and various ablations of LaSR. Our experiments are fully reproducible, and we plan on releasing our logs with our codebase, which should highlight more details.
>
> > **Could you clarify what constitutes an "iteration" ... There are discrepancies between ... operations will be used?**
>
> An iteration is one full run of the SR cycle, and the concept crossover steps (this is best expressed in Algorithm 1). $10^6$ in Figure 1 is the total number of *operation calls per iteration*. As mentioned in Appendix 3.1, LaSR makes $60000$ operation calls per iteration, $60000$ calls/iteration * $40$ iterations = $2.4 \times 10^6$ calls. We have updated Figure 1 to say “$10^6$ operations” instead of “$10^6$ iterations.” Thank you for pointing this out!
>
> We cannot run LaSR (or PySR) for $10^6$ iterations, as that would take roughly $10^6$ to $10^7$ minutes on an 8 x NVIDIA A100-80GB server node. As mentioned in Appendix 3.1, assuming $p=0.01$, we will make $60000$ calls per iteration so $0.01 * 60000 = 600$ LLM operation calls (roughly) per iteration.
>
> >  **For the data leakage validation, ...  black-box datasets in SRBench?**
>
> Great question! We wanted to compare PySR’s performance and LaSR’s performance on a set of unknown problems with ground truth equations available, which is why we chose to procedurally generate our own equations. The black box datasets in SRBench do not have ground truth equations.
>
> > **Could you provide a computation time comparison?**
>
> Our goal is to present experimental results that can be replicated reliably by the broader community. As such, we root our experiments in the number of iterations instead of wall time.
>
> This is because the wall time is heavily dependent on the LLM backend, the number of tokens each query uses, and the network quality. For instance, for `gpt-3.5-turbo` we cannot reliably measure the runtime as the backend is slower during periods of high traffic. Even for local models, vLLM's query scheduling is non-deterministic, so the wall time can be substantially different across experiments.
>
> Regardless, we were measuring around one second per query using vLLM. Since we make 60k calls per iteration (Appendix A.3.1) the amount of time for p=1% would be around 600 seconds / 10 minutes per iteration. PySR roughly takes 30 seconds per iteration. We expect performance to improve as we increase CPU/GPU parallelism, run LaSR on dedicated backend hardware, and as the hardware and software to run LLM inference improves.
>
> > **(...) examples of "hints" in Feynman equations?**
>
> As mentioned in Section 4.4, we use the title of the chapter the equation appears in in the Feynman Lectures on Physics. For instance, the Shockley diode equation is introduced in the “Semiconductors” chapter of the Feynman lectures. Hence, the hint would be “Semiconductors.”
>
> > **Based on Table 2, (...) Could you explain why higher values were not explored?**
>
> As we mention in the limitations sections, our evaluation was constrained by our compute budgets for LLMs and search. Whether these trends hold for higher compute regimes remains to be seen.

---

> > ### Comment · Reviewer_1Xi9 · 2024-08-12
> > **Response to the authors**
> >
> > Thank you for your responses. Some of my question have been answered. But I still have some concerns including:
> >
> > 1. For the lack of experiments on SRBench black-box datasets, the authors mention that they wanted to experiment on datasets with known ground-truth equations. However, first, the authors have not reported on the function recovery for this dataset. More importantly, LaSR is eventually compared based on fitness (R2) in sec 4.5 and table 3, which could have been the case for SRBench black-box datasets.
> >
> > 2. I understand that using LLMs can be time-consuming, and this is in no ways a downside. However, I find comparing LaSR and PySR on same number of iterations (and with only 40 or 400 iterations in Fig.3 and table 3) not fair, since they require very different computation and wall-clock time. Also, since PySR is a GP-based method and lacks prior knowledge, it would be reasonable for them to need more number of iterations to converge to the same level of performance. So it would be more reasonable to report the results after all the methods have converged (or at least same computation/wall-clock time), and not necessarily at the same iteration.
> >
> > 3. In the current results shown in Fig 3, without the additional memorization hints (variable name, hints, or mention of physical concepts), the improvement of LaSR (which in this ablation could be potentially attributed to the LLM role in learning good vs. bad forms) looks marginal. Also, this plot shows the results on MSE and not the exact match as reported in table 1. So it is not shown that how many equation are discovered out of 100 in this condition. Based on figure 3, PySR has solved ~41 equations after 40 iterations on MSE metric but based on table 1, it has solved 59 equations based on exact match. This discrepancy in the metrics and the final results of LaSR in this condition should be explained.
> >
> > Other concerns and comments:
> >
> > As an additional note, the authors mention that in table 1, the 7 improved equations are consistent across all runs. As both initialization of GP methods and the responses of LLMs involve some level of variance, this would increase the concern that these 7 enhanced equations could have in fact obtained due to the memorization rather than pure reasoning and learning on the equation forms.
> >
> > The observation on the form of discovered equations is interesting and worth discussing, however, not generating the exact known form of equations does not guarantee that the model has not used its internal memorization as a part of its inference.  I would suggest authors to include a discussion in the paper on the results of LaSR without any variable name and even a notion of "physical concepts".

---

> ### Author Response · Authors · 2024-08-13
>
> Thank you for engaging with us and helping us improve this work! Apolgies for the delay! PySR experiments took a while. We respond to your points inline:
>
> > ground truth equations
>
> We’re happy to include the LaSR’s discovered equation and the ground truth equations in the broader artifact release. We did not report on exact match for ground truth equations because neither the equations discovered by PySR nor those by LaSR were close enough to regard as “exact matches.” Instead, we relied on correlation metrics used in the SR community specifically for this circumstance. As shown in Table 3, we find that LaSR’s equations are far closer to the ground truth data as compared to PySR’s equations.
>
> > Comparison on wall time instead of # iterations
>
> In our opinion, comparing w.r.t # iterations is a fair experimental strategy for several reasons:
>
> 1. In practice, evaluating an equation itself is more expensive than the LLM queries, and thus the # iterations becomes directly correlated to time. For example, you might want to use the discovered expression inside a scientific simulation, which can be very expensive to evaluate. For such problems, the wall clock time is a function of the # iterations alone.
>
> 2. As mentioned in our original rebuttal, evaluating on wall clock time is less reliable and reproducible than evaluating on the # iterations.
>
> 3. Regardless, LaSR’s experiments took roughly 10 hours on an 8xA100 80GB server node. We ran PySR for 10 hours (capped at 1 million iterations) on the Feynman equations dataset and report results here:
>
>    | Feynman Equations | PySR ($1000000$ Iterations, 10 hour timeout) | LaSR ($40$ iterations) |
>    | ----------------- | -------------------------------------------- | ---------------------- |
>    | Exact Solve       | 59 + 3/100                                   | 59 + 7/100             |
>
>    Overall, we find that, despite running PySR substantially longer than LaSR, LaSR still substantially outperforms PySR. This is because PySR, like other evolutionary algorithms, is susceptible to falling in local minima and converges to this local minima pretty fast. This is why supplementing such “local” search strategies with LLM guidance is useful.
>
>
> > The improvement of LaSR looks marginal
>
> In scientific discovery, finding even one additional equation can be highly valuable. We are encouraged by the fact that LaSR's ablations still outperform the current state-of-the-art algorithm (PySR) under the same conditions.
>
> > Differing  performance for MSE metric and exact match metric
>
> Exact match is rooted in the symbolic equations sketch discovered, while MSE is based on how well the symbolic equation fits to the dataset. The latter is susceptible to optimization errors, especially since these optimization methods do not guarantee finding a global minima (we use BFGS for all experiments). Also, we evaluate on a challenging version of the Feynman dataset where we add random noise to each equation. This further exacerbates parameter fitting errors.
>
> The MSE threshold metric is necessary for gauging performance per iteration because evolutionary algorithms constantly evolve a pool of candidates and the MSE threshold allows localizing when the best performing candidates start appearing in the candidate pool.
>
> > Concerns about memorization / equation consistency
>
> There is a slight miscommunication here. LaSR consistently discovers solution equations for these seven problems. However, the discovered equations do not necessarily follow the same format. For instance, Equation 53 in the Feynman equations dataset defines the heat flow from a point source (an inverse square law with ground truth formulation: $h = \frac{P}{4\pi R^2}$). LaSR finds the following solutions in successive runs with the same hyperparameters:
>
> | Discovered Equation (LaSR - Llama 3b 1%; same params)        | Equation Loss      | Equation Complexity |
> | ------------------------------------------------------------ | ------------------ | ------------------- |
> | $h = \frac{2.8841133}{0.000219 - \left( r \cdot \left(\frac{-36.240696 - (\sin(r) \cdot 0.002057)}{P}\right) \cdot r \right)}$ | $2 \times 10^{-6}$ | $16$                |
> | $h = \frac{P}{r \cdot (r \cdot 17.252695 - 0.001495)} \cdot 1.372847 $ | $2 \times 10^{-6}$ | $11$                |
>
> Both equations fit well to the underlying dataset and reduce to the ground truth. Yet, despite using the same hyperparameters, the functional forms exhibit sharp differences.
>
> > The observation ... part of its inference
>
> We are encouraged that LaSR's equations differ significantly in format from the known ground truth equations. It is highly unlikely that the discovered equations are available verbatim online, further reinforcing our hypothesis that LaSR’s performance is not simply the result of  regurgitated memorized responses.
>
> > Discussion of LaSR ...  physical concepts
>
> We’re happy to add relevant discussions to the manuscript. Thank you for the excellent suggestions!

---

### Author Rebuttal · Authors · 2024-08-07

We’d like to thank the reviewers for their thoughtful comments and suggestions. We've incorporated many suggested changes in our manuscript and will update the PDF whenenver the portal opens up next.

Multiple reviewers expressed concerns about data leakage from LLMs. While this is a valid concern for all LLM-based approaches, we have strong evidence against data leakage:

- **Synthetic Data experiments:** The synthetic dataset contains 41 novel and challenging equations that do not exist in any LLM training dataset. We find that LaSR outperforms PySR on this dataset; strongly reinforcing that LaSR’s performance gains are not rooted in memorization or data leakage.
- **Ablation without common identifiers:** Figure 3 (Left) showcases that LaSR, in the absence of common identifiers like variable names, still outperforms PySR.
- **Evidence from qualitative evaluation:** In Appendix A.4.1, we analyze the equations discovered by PySR and LaSR from data corresponding to Coulomb's law. If LaSR’s performance was rooted in regurgitating memorized data, LaSR’s equation would correspond to the common formatting of Coulomb's law ($F=\frac{1}{4\pi\epsilon_0} \frac{q_1 q_2}{r^2}$). Instead, we observe that LaSR’s discovered equation displays a very different format, and needs at least five simplification steps to transform to the commonly seen equation.

We elaborate on these points, and address the reviewers' other concerns, in the individual responses below.

---

### Decision · Program_Chairs · 2024-09-25

**Decision:**

Accept (poster)

**Comment:**

The Reviewers agree that the method is well-motivated, and aligns well with the goal of developing a system that reflects how humans discover useful equations. Unlike prior work that uses LLMs to generate expressions for symbolic regression, the new approach, LaSR, is integrated w/ an off-the-shelf classical evolutionary SR method, so it is able to exploit the advantages of both paradigms. The method works as expected, performing well empirically, and the paper can be accepted along these lines.

The main concern with this kind of paper is information leakage, since the LLM has certainly been exposed to all the Feynman equations in its training data. The authors have taken substantial steps to address this concern, including: demonstrating the approach on a previously unpublished set of synthetically generated equations, demonstrating that the approach discovers different functional forms of the same equation over different runs, and running ablations w/ a priori knowledge removed.

This does not completely eliminate concerns of data leakage, but I would argue that completely eliminating this concern is not possible at this time. With LLM-based approaches, even knowledge of black-box problems and best known solutions so far could bias the model; and, at a higher level, knowledge of a stochastic synthetic equation generator could have a similar impact. I would argue that existing methods including PySR implicitly have tons of prior knowledge baked in, since they were developed with the benchmark in mind of rediscovering known physical equations. Overall, establishing completely "fair" comparison of SR methods remains an open problem that is outside the scope of this work. That said, I have four recommendations for clarifying the contribution:

(1) In the paper, articulate the challenges of "fair" comparison as fully as possible.

(2) I encourage the authors to follow the reviewer advice to include results on black-box SRBench problems in the final version. Although the authors say they are interested in scenarios where the ground truth is known, in a real application of such a method there would be no ground truth. These experiments could also show a "real new discovery" if they find equations on the Pareto front of known best equations. Similarly, if LaSR leads to another "new discovery" before the final version, e.g., in scaling laws as the authors have suggested, I encourage them to add it to the paper.

(3) The result that different trials produce different functional forms of the same equation should be clearly highlighted in the paper, e.g., in a table showing differing discovered solutions across different trials for multiple problems.

(4) Clearly explain the precise motivation and construction of the synthetic domain.

One interesting result in the paper was that the LLM could be used to generate only 1% of expressions and still yield notable benefits. I would really like to see the other extreme, i.e., how well does the method do at 100% LLM? If it's possible to add these experiments to the final version, it would better contextualize the method compared to existing LLM-only methods, e.g., LMX [1], the first evolutionary approach to use LLMs to equations for SR. The Related Work section would generally benefit from discussions of such earlier approaches. As is, the reader could reach the incorrect conclusion that LaSR is the first method to generate SR equations w/ LLMs.

[1] Meyerson et al. "Language Model Crossover: Variation through Few-Shot Prompting" arXiv:2302.12170